# A crop yield change emulator for use in GCAM and similar models: Persephone v1.0

Abigail Snyder[1], Katherine V. Calvin[1], Meridel Phillips[2, 3], and Alex C. Ruane[3]

[1]Joint Global Change Research Institute, Pacific Northwest National Laboratory, College Park, MD 20740
[2]Columbia University Earth Institute Center for Climate Systems Research, New York, NY, USA
[3]NASA Goddard Institute for Space Studies, New York, NY, USA

**Correspondence:** Abigail Snyder (abigail.snyder@pnnl.gov)

**Abstract.** Future changes in Earth system state will impact agricultural yields and, through these changed yields, can have profound impacts on the global economy. Global gridded crop models estimate the influence of these Earth system changes on future crop yields, but are often too computationally intensive to dynamically couple into global multi-sector economic models, such as GCAM and other similar-in-scale models. Yet, generalizing a faster site-specific crop model's results to be used globally will introduce inaccuracies, and the question of which model to use is unclear given the wide variation in yield response across crop models. To examine the feedback loop among socioeconomics, Earth system changes, and crop yield changes, rapidly generated yield responses with some quantification of crop response uncertainty are desirable. The Persephone v1.0 response functions presented in this work are based on the Agricultural Model Intercomparison and Improvement Project (AgMIP) Coordinated Climate-Crop Modeling Project (C3MP) sensitivity test data set and are focused on providing the Global Change Assessment Model (GCAM) and similar models with a tractable number of rapid to evaluate, dynamic yield response functions corresponding to a range of the yield response sensitivities seen in the C3MP data set. With the Persephone response functions, a new variety of agricultural impact experiments will be open to GCAM and other economic models; for example, examining the economic impacts of a multi-year drought in a key agricultural region and how economic changes in response to the drought can, in turn, impact the drought.

## 1 Introduction

Agricultural yields are susceptible to changes in temperature, precipitation, growing season length, $CO_2$ concentrations, and other Earth system factors. While both the nature of the future climate and its impact on agricultural yields are uncertain (Rosenzweig et al., 2014; Pirttioja et al., 2015; Fronzek et al., 2018; Asseng et al., 2013, 2015; Martre et al., 2015; Lobell, 2013), it is clear that there is potential for identifying the important effects on agriculture and, in turn, the economic state of the world at large. The global multi-sector economic model Global Change Assessment Model (GCAM)[1] (Kyle et al.,

---

[1]Model and documentation available at https://github.com/JGCRI/gcam-core, http://jgcri.github.io/gcam-doc/toc.html

2011; Wise et al., 2014; Calvin et al., 2019; Hartin et al., 2015) and other similar-in-scale models (Nelson et al., 2014) are ideal for understanding the far reaching impacts of this climate-agriculture-economic cycle, but rely on external projections of agricultural yields to quantify these effects (Figure 1, panel A). This asynchronous process results in inconsistencies between the economic and biophysical world, and overlooks feedbacks and unintended consequences as the future shifts (Ruane et al., 2017).

Several modeling groups, including the GCAM model development team, are interested in explicitly modeling and understanding bidirectional feedbacks between the Earth and the human systems (e.g. Figure 1, panel C). Agriculture is one important pathway (of many) through which these systems directly interact. A prime example would be to study the impacts of a multi-year drought in a key agricultural region. The drought would affect yields, which would affect the agricultural supply to the global economic market. In a model like GCAM, this would lead to price changes and shifting land to more profitable crops. The new spatial distribution of agricultural land would change land related emissions, which will in turn affect climate and therefore yields moving forward. Being able to model each component of this process and the interactions among them is key to considering important questions like this one.

Currently, GCAM operates on a five year time step and is coupled with a physical Earth system emulator, Hector (Hartin et al., 2015) (as in Figure 1, panels A and B), to explore global change questions in rapid enough evaluation times to allow for large numbers of simulations to be analyzed as part of a wide range of experiments. GCAM is a recursive dynamic partial equilibrium model that is calibrated to a historical base year of 2010 and used to simulate forward in time by incorporating changes in quantities such as population, GDP, and technology to produce outputs that include land, water, and energy use as well as emissions and commodity prices. For agricultural production in GCAM, yield change trends representing generally positive change assumptions over time due to *non-climate* factors (changes in management, new seed genetics, new technologies, use of chemicals/fertilizers, adaptation, etc.) are used to calculate the profitability of a crop-irrigation-fertilizer combination in each of 384 GCAM land units at each time step based on the global crop price. This profitability determines land allocated to each crop, and the combination of exogenous yields and land allocation gives production of each crop-irrigation-fertilizer combination such that global supply and global demand are met on each timestep. The details of this allocation are provided in Kyle et al. (2011); Wise et al. (2014); Calvin et al. (2019). Shifting land allocation among different crop-irrigation-fertilizer combinations leads to a degree of endogenous yield intensification within GCAM.

Past agricultural impacts studies using GCAM (Calvin and Fisher-Vanden, 2017) have focused on using outputs of global gridded crop model (GGCM) studies (e.g., Rosenzweig et al., 2014; Elliott et al., 2014; Müller et al., 2017) in a strictly feed-forward way (Figure 1, panel A). Direct coupling of a GGCM to GCAM would result in a computationally expensive modeling framework, limiting the number of simulations that could be performed. Yet, large ensembles of simulations are necessary to explore and understand future response options, so there is great need for a computationally efficient model that could explore the uncertainty space. While GCAM is already coupled to a simple climate model, Hector (Hartin et al., 2015), this coupling is one-way: emissions are passed to the climate model, but to date dynamic feedbacks between climate and humans at each timestep are missing. In this paper, we describe the first version of Persephone (v1.0), a simple representation of mean agricultural response and uncertainty to future climate that can be incorporated into GCAM and similar models.

Further detail of the desired studies this yield change emulator would be used for are given in Section 2.1 and discussed at length in Ruane et al. (2017).

An ideal solution to the computational expense of coupling a GGCM to GCAM is a yield response emulator, which uses past crop yield model runs to predict what the model *would* have done under different conditions, had it been run. However, previous

work in this area has been restricted to either emulating crop model results under fixed [$CO_2$]-temperature pathways such as the various RCPs (Oyebamiji et al., 2015; Blanc, 2017; Ostberg et al., 2018) or building statistical models from empirical and historical data (Lobell, 2013; Moore et al., 2017; Mistry, 2017; Mistry et al., 2017). While an emulator trained on RCP-driven scenarios can be used to estimate yield change in any future climate, the RCPs only span a subset of possible future climates. In particular, should one want to consider the impacts of [$CO_2$]-temperature pathways that substantially differ from the RCPs,

these emulators would face the difficult task of predicting yield changes outside of the conditions of the training data. Statistical models of empirical and historical data also must predict yield changes in response to future climate outside of the conditions of the training data, especially in response to large [$CO_2$] increases. Substantial departure from the RCPs and historical values of [$CO_2$] is very possible in the bidirectional coupled human-earth system applications outlined above and an emulator equipped to handle that is desirable. Finally, many of these past studies have lacked a way to capture aspects of uncertainty that would

be useful for the GCAM bidirectional feedback experiments described in Section 2.1.

The Agricultural Model Intercomparison and Improvement Project (AgMIP) (Rosenzweig et al., 2013) took steps to begin addressing these issues with the Coordinated Climate-Crop Modeling Project (C3MP), a modeling study specifically designed to, among other things, provide the data necessary to develop a flexible and dynamic crop yield emulator (Ruane et al., 2014; McDermid et al., 2015). C3MP invited point-based crop modelers from across the AgMIP community to simulate

their calibrated agricultural system's response to 99 sensitivity tests in which 1980-2009 baseline climate data were modified to synthesize changes in mean carbon dioxide concentration ([$CO_2$]), temperature, and precipitation. The 99 Carbon-Temperature-Precipitation (denoted CTW, W for Water rather than P for Precipitation) tests that make up the C3MP protocol were selected using a Latin hypercube to ensure that future scenarios through the end of the 21st century, including all RCPs, fall within the training model simulation data over the vast majority of agricultural lands (Ruane et al., 2014). The full space of CTW changes

that these 99 tests represent is: 330-900 ppm global [$CO_2$], -1°C to +8°C from local baseline temperature, and -50% to +50% from local baseline precipitation (applied as a multiplicative factor). A particular CTW perturbation could be associated with a specific time slice, for example the 2050s climate changes from a given Earth System Model (ESM) RCP4.5 projection, or from a climate condition generated within GCAM as a result of interactions between socioeconomic development and the natural environment. Finally, the C3MP study featured broad spatial coverage (albeit not uniform) of a wider variety of crop models,

crops, and management practices than has been incorporated into past GGCM or emulator work. More than 50 participating crop modelers helped C3MP record yield response simulation results from a total of 1135 sites, differing by location, crop species, cultivars, crop model, farm management, etc.

The Persephone framework presented in this work is designed to develop yield response functions to CTW changes from a given data set. The Persephone V1.0 response functions, based on the C3MP data set, provide a computationally inexpensive

estimate of the change in agricultural yield due to a change in the Earth system, and make use of the promising data relating

yield changes to CTW changes collected in C3MP. Specifically, we present biologically reasonable response functions that are rapid-to-evaluate and more dynamic than past options for incorporating crop responses into models like GCAM. We strictly considered responses to long term Earth system changes. The C3MP results or other appropriate data sets could be further used to examine the effect of inter-annual variability on yields in Persephone V2.0 and beyond, although this would require additional complexities in seasonal yield variations that are largely averaged out in long-term trends. The response functions also represent the uncertainty in yield response across crop models in the C3MP data set to a given change in local Earth system state, for use in three types of agricultural impacts studies with GCAM:

1. A partially coupled, feed forward study (Figure 1, Panel B) similar to methodology in Ruane et al. (2018). A future climate time series of interest (a non-traditional RCP, climate stabilization level, or hypothetical drought, for instance) is input to the yield response functions, returning yield changes. These yield changes are applied as multipliers to GCAM input files and GCAM is run forward for the entire time period of interest in order to trace the broad impacts on energy, water, and land use of the future climate time series. In this type of study, we only capture the implications of climate for human systems.

2. A fully coupled feedback loop that updates on every model timestep to understand how societal pressures drive environmental impacts which in turn create or reduce societal pressures (Figure 1, Panel C). In this case, the yield changes must be calculated very quickly in order to evaluate on each step and interact with GCAM. In this type of study, we can capture the effects of humans on climate and climate on humans, simultaneously.

3. Joint climate-crop uncertainty studies of the above two experiments. For tractability, the GCAM development team specifically seeks a mean response function as well as two additional response functions that represent a range of yield response uncertainty. Persephone also stores the full predictive distributions of yield changes for any given CTW change that these three response functions span. If a user desires a different representation of uncertainty, the distribution may be sampled.

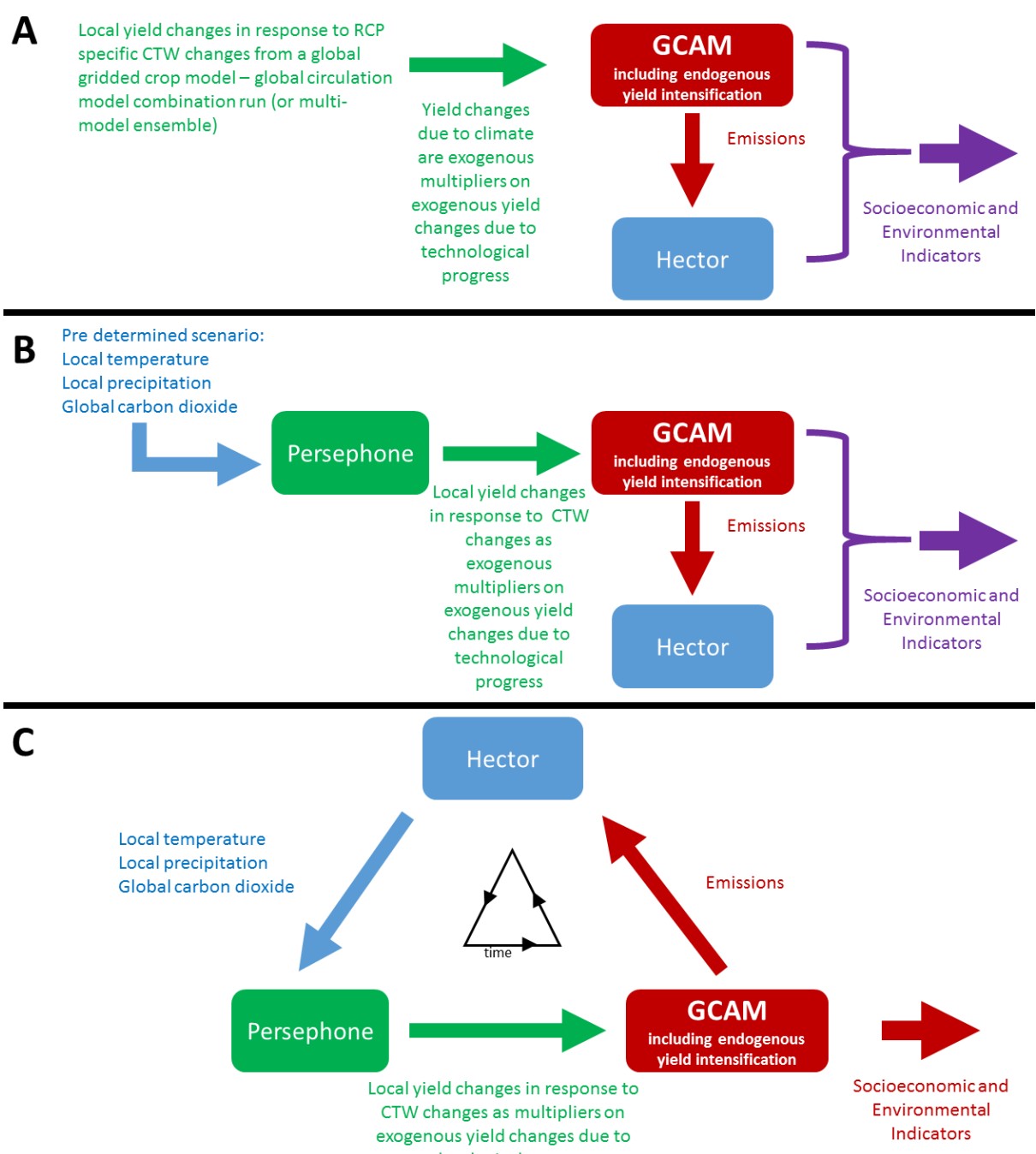

**Figure 1.** The current method for incorporating agricultural impacts into GCAM and two experimental designs for using Persephone v1.0 with GCAM. Panel A: The current method for incorporating yield changes from a global gridded crop model into GCAM. Panel B: A partially coupled, feed forward study incorporating yield changes from a predetermined climate scenario into GCAM. Panel C: A fully coupled feedback loop that iteratively updates agricultural yield impacts.

## 2 Methods

### 2.1 C3MP dataset

Full details of the C3MP protocols, design, and the location output archive can be found in Ruane et al. (2014); McDermid et al. (2015). Here, we highlight some of the key features of the data set and outline our processing of C3MP data for using the Persephone framework to train V1.0 response functions with the Persephone framework.

C3MP recorded yield response simulation results from a total of 1135 sites (differing by location, crops, crop model, management, etc) for each of 99 CTW sensitivity tests designed to cover a range of CTW changes that most future climates would fall into. For each site, each CTW test is applied to change a local time series of weather data from 1980-2009 and then the crop model is run to produce 30 years of impacted yields for the CTW test, which are then averaged.

The C3MP design resulted in a wider range of crops than had been previously sampled in a coordinated agricultural modeling study. We separate the C3MP data into 25 different production groups for training in the Persephone framework to create V1.0 response functions. Twenty-four of the 25 groups for this paper are collections of sites corresponding to different crop-irrigation-latitude combinations: irrigated and rainfed versions of six key crops (maize, rice, wheat, soybeans, a C3-photosynthesis average, and a C4-photosynthesis average ), based on sites at the extended tropics (30°S to 30°N) and the mid-latitudes (30- 70°S, 30- 70°N) (see Section 2.1.1 for more details on spatial scales). It is also noteworthy that the majority of C3MP sites had high rates of fertilizer application, even in the extended tropics. These six crop groups were chosen because most IAMs already have experience incorporating such impacts from previous AgMIP exercises (e.g., Ruane et al. (2017); Calvin and Fisher-Vanden (2017); Nelson et al. (2014); Wiebe et al. (2015); Ruane et al. (2018)), they cover the major agricultural commodities globally, and they offer additional benchmarks for evaluating emulator success. In particular, the C3-photosynthesis production groups represent an average response of a very wide range of C3 crops, including wheat, rice, and soybeans. The C4-photosynthesis average is similarly defined, with sugarcane considered separately. The 25th production group is rainfed sugarcane in the extended tropics: no sugarcane sites outside of 30°S to 30°N were submitted to C3MP and only one irrigated sugarcane site was submitted.

We cull the 1135 contributed C3MP output datasets according to a range of criteria:

1. Sites simulated with notably older versions of crop models are eliminated. We thus eliminated uses of the DSSAT crop model v3 (and prior), given that important updates in crop physiology were added in version 4 (Jones et al., 2003).

2. Site simulations that exclude $CO_2$ fertilization responses, a fundamental variable examined here, were eliminated. We thus eliminated the SarraH-Hv32 crop model (primarily millet and sorghum sites in West Africa).

3. When C3MP modelers provided simulation sets that were identical other than the use of local weather data or AgMERRA climate forcing data (Ruane et al., 2015)), we used only the local dataset to avoid double counting. AgMERRA was provided for all datasets given frequent data gaps and governmental restrictions (Ruane et al., 2014).

These steps together eliminate more than 550 of the C3MP sites. Finally, for each production group, outliers are statistically identified and eliminated (Davies and Gather, 1993; Bond-Lamberty et al., 2014), in addition to those previously identified by the C3MP steering team. A total of 575 unique sites remain after culling, maps of which are included in Figure 2. These remaining sites cover 43 countries, 85 models, and 17 crop species. More than half of the C3MP sites have been eliminated, but this still results in a larger number of diverse sites, models, and crop species performing coordinated sensitivity tests than in any previous study (Asseng et al., 2013; Pirttioja et al., 2015; Fronzek et al., 2018). Since C3MP, the AgMIP-Wheat team has conducted an extensive analysis of temperature response at 30 wheat sites with 30 models (Asseng et al., 2015), but this only captures one of the CTW dimensions.

### 2.1.1 Known caveats of the C3MP data set

Additional discussion of the C3MP data set in the context of other AgMIP modeling efforts is presented in Ruane et al. (2017). One relevant point to this work is that, while C3MP spatial coverage is not spatially uniform or production-weighted for any of the crops under consideration, sites for many of the major production regions are represented for each crop (Figure 2). A major advantage of using site-specific crop models run voluntarily by experts is that the individual baseline runs at each site have been configured against local information in the historical period. However, the application of crop yield response from these sites to estimate response in any given grid cell with temperature and precipitation data is imperfect by its methodological nature. Yet this extension is necessary for use with GCAM: gridded yield changes for a subset of crops must be aggregated and converted to yield impact multipliers for each GCAM commodity in each land unit, defined as water basins in GCAM (Calvin et al., 2019).

Given the size and details of the C3MP data set, production groups were formed based on two latitude zones as a way to account for baseline local temperature (which is important in addition to the change from local temperature) without having to eliminate the many valid C3MP sites that could not report local weather data due to data gaps or local government restrictions. As this breakdown already results in some production groups with small sample sizes (see Table 1 and Section 3.1.1), further spatial disaggregation of production group is unjustified in this data set. While this means there will be limited spatial granularity in yield response *functions*, there can still be appreciable spatial granularity in yield *changes* due to variation in the gridded fields of temperature and precipitation changes. Future data sets with more comprehensive spatial coverage than the C3MP data may be used rather to create V2.0 response functions.

The site-specific percent change in yield from the 1980-2009 baseline yield is the dependent variable used to train our emulator (see Section 2.2). Baseline yields differ widely across the C3MP archive due to regional and system differences, however the percent change in yield from baseline is more consistent across sites for each CTW. Further, by training on change in yield rather than yield, we are able to introduce additional, scientifically grounded constraints to the functional forms we fit (Equations (4) - (6)). However, no baseline simulation was requested under the C3MP protocols. Therefore, for each individual set of output yields corresponding to each of the 575 simulation sites, we perform ordinary least squares regression for eight different functional forms relating the site-specific output yield to the input CTW values and select the best performing regression to estimate baseline yield (details in Appendix B, Equations (B1)-(B8)).

It is also worth noting that the C3MP experimental protocols (Ruane et al., 2014; McDermid et al., 2015) do not account for changing growing seasons, either through changes of within season distribution of temperature and rainfall or in the possible autonomous adaptation of farmers to shift planting and harvest dates. Ruane et al. (2014) showed that within season distribution changes had a small effect and the possible shift in planting and harvest dates are a topic of adaptation. Modeling autonomous adaptation behaviors is a challenging area for coordinated agricultural efforts and is only beginning to be addressed in coordinated sensitivity intercomparison studies as a scenario option, with no publicly available data sets at this time.

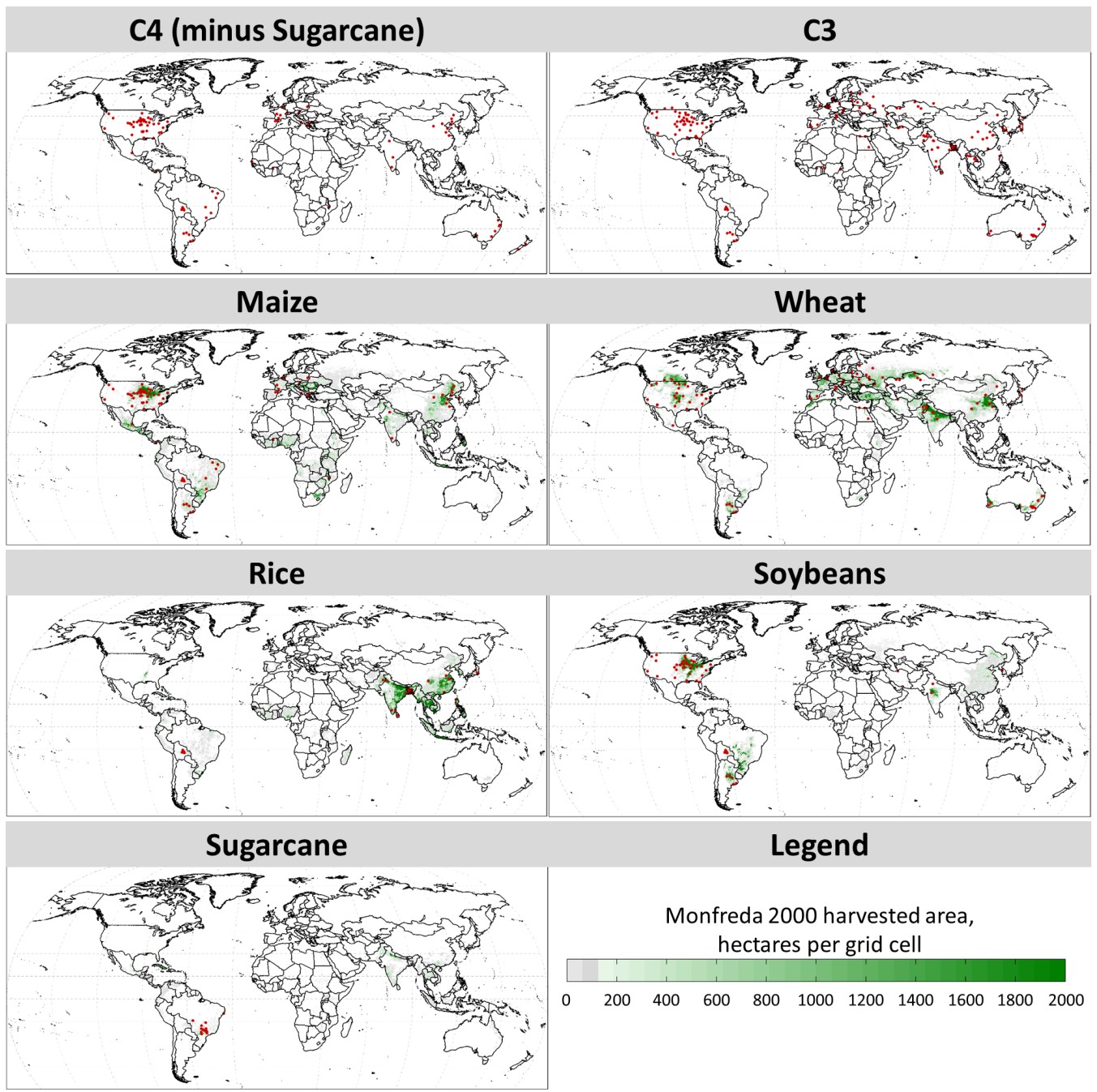

**Figure 2.** Maps of the C3MP data set culled sites. Each site represents a site-specific model of a single crop, with differing management practices. The sites are overlaid on Monfreda et al. (2008) harvested area data, except for the C3 and C4 averages.

## 2.2 Emulation

The majority of past agricultural yield emulator work has used ordinary least squares regression to estimate coefficients of functional forms. Given a set of predictors, $\mathbf{x}$, and given a particular value of the predictors $\mathbf{x}_i$ with corresponding training data $y_i$, an emulator would be some linear-in-parameters function $f(\mathbf{x})$ that returns an emulated value $f(\mathbf{x}_i)$ for comparison with $y_i$. Ordinary least squares regression requires that residuals $r_i = y_i - f(\mathbf{x}_i) \sim N(0, \sigma^2)$ for all $i$ (e.g., Williams and Rasmussen, 2006, Section 2.1.1). A key requirement is that $\sigma$ is a constant value across all $i$.

Figure 3 displays the spread of yield responses across sites for each CTW test for one production group, rainfed soybeans between 30- 70°S, 30- 70°N (the mid-latitudes). A successful emulator will produce the mean response (Figure 3, black dots) across sites for each CTW. Therefore examining the spread of the individual site yield changes about the mean yield gives some sense of the behavior of residuals in the most successful emulation case.The spread of yield change across sites relative to the mean response is different for each CTW test and appears to change in a systematic way - larger magnitude changes in yield are correlated with greater spread across sites. In light of this, a classic, ordinary least squares regression is not an appropriate approach for this emulator. We also desire more than just the mean response: we desire a measure of how this variation of site responses changes with CTW. With these considerations in mind, we take a slightly different approach to creating the Persephone V1.0 response functions, working from texts such as Gelman et al. (2013); Sivia and Skilling (2006); McElreath (2016).

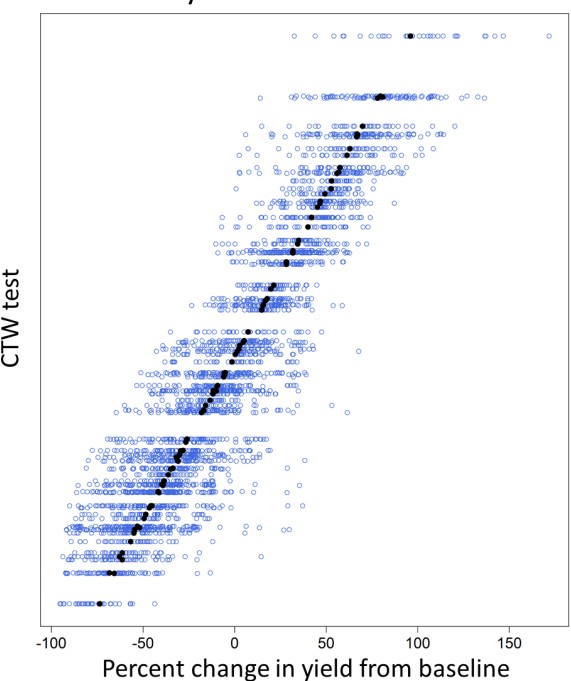

**Figure 3.** A plot of the percent yield change at each rainfed soybeans in the mid-latitudes site (blue points) for each CTW test (each horizontal line of points is a different test). The black dot for each test represents the mean response across the sites for that test.

We create the Persephone V1.0 response functions to emulate the mean yield response and two additional yield response scenarios spanning a range of individual site responses. For a given production group (crop - irrigation - latitude zone combination), we collect the data for the 99 CTW tests for each of K C3MP simulation sets drawn from the culled-down archive. In other words, for each of 99 CTW combinations, there exist $K$ 30-year average yield percent changes from the baseline (no changes

in CTW) for a group. This ensemble of $99K$ yield changes is used to calculate the posterior densities for every parameter of $\mu_{CTW}$ and $\sigma_{CTW}$ in the model defined by Equations (1) - (7) according to Bayes' theorem ($posterior \propto likelihood \times prior$). From the posteriors, the *maximum a posteriori* (MAP) estimates of parameters, the most plausible value for each parameter given both the model being used and the training data, is returned.

We define our likelihood as a normal distribution with mean $\mu_{CTW}$ and variance $\sigma^2_{CTW}$:

$$\Delta Y^{emulated}_{CTW} \sim N(\mu_{CTW}, \sigma^2_{CTW}) \tag{1}$$

For a production group with site-specific yield responses that are normally distributed for each CTW value, $\mu_{CTW}$ is the mean response across sites for that CTW value (the black points in Figure 3), and $\sigma^2_{CTW}$ is a measure of agreement (or disagreement) of responses across sites for that CTW value. We present results for our most broadly optimal mean and variance functional form combination in this paper, and present the details of our selection criteria among the different functional forms in the

15 Appendix.

To have unitless coefficients in our emulator, all predictor variables are standardized. Defining the collection of 99 T changes sampled by C3MP as $T_{C3MP}$, the collection of precipitation changes as $W_{C3MP}$, and the collection of $CO_2$ concentrations as $C_{C3MP}$, we have:

$$\Delta T = \frac{T - T_{baseline}}{sd(T_{C3MP})}$$
$$\Delta W = \frac{W - W_{baseline}}{sd(W_{C3MP})} \tag{2}$$
$$\Delta C = \frac{C - C_{baseline}}{sd(C_{C3MP})}$$

$T_{baseline}$ is a change of $0°$ C from baseline, $W_{baseline}$ is a $0\%$ change in precipitation from baseline, and $C_{baseline}$ is 360ppm. Plugging these baseline values into Equation (2) returns $\Delta T_{baseline} = \Delta W_{baseline} = \Delta C_{baseline} = 0$, as one would expect.

We exploit the fact that we are emulating change in yield (and not yield) and the fact that $\Delta T_{baseline} = \Delta W_{baseline} = \Delta C_{baseline} = 0$ in constructing Equations (4)-(7), which relate the mean and standard deviation of the likelihood in Equation (1) to our unitless predictor values $\Delta C, \Delta T, \Delta W$. By definition, percentage change in yield in response to no change in CTW

is $0\%$ at baseline for *every* individual C3MP site. This implies that both mean and variance at baseline are 0 for all production groups, and we must construct the Persephone response functions to reflect this, independent of the estimated baseline yield at each site:

$$\mu_{baseline} = 0$$
$$\sigma^2_{baseline} = 0 \tag{3}$$

Implementing this constraint for the mean is straightforward. Any functional form representation of $\mu_{CTW}$ that does

not include a constant parameter $a_0$ will force $\mu_{baseline} = 0\%$ yield change precisely because $\Delta T_{baseline} = \Delta W_{baseline} = \Delta C_{baseline} = 0$.

$$\begin{aligned}
\mu_{CTW} = {} & a_1\Delta T + a_2(\Delta T)^2 + a_3\Delta W + a_4(\Delta W)^2 + a_5\Delta C + a_6(\Delta C)^2 + a_7\Delta T\Delta W + a_8\Delta T\Delta C + a_9\Delta W\Delta C \\
& + a_{10}\Delta T\Delta W\Delta C + a_{11}(\Delta T)^2\Delta W + a_{12}(\Delta T)^2\Delta C + a_{13}\Delta T(\Delta W)^2 + a_{14}\Delta T(\Delta C)^2 + a_{15}(\Delta W)^2\Delta C \\
& + a_{16}\Delta W(\Delta C)^2 + a_{17}(\Delta T)^3 + a_{18}(\Delta W)^3 + a_{19}(\Delta C)^3
\end{aligned} \tag{4}$$

Constraining the variance to be 0 at baseline as in Equation (3) should be equally easy by simply not considering any functional form that includes a constant parameter. However, this approach leads to numerical stability issues when estimating

parameters. Therefore, we estimate the variance using the following functional form:

$$\sigma^2_{CTW} = \left(b_0 + b_1\Delta T + b_2(\Delta T)^2 + b_3\Delta W + b_4(\Delta W)^2 + b_5\Delta C + b_6(\Delta C)^2 + b_7\Delta T\Delta W + b_8\Delta T\Delta C + b_9\Delta W\Delta C\right)^2 \tag{5}$$

This results in the following functional form representation for the standard deviation:

$$\sigma_{CTW} = +\sqrt{\sigma_{CTW}^2}$$
$$= |b_0 + b_1\Delta T + b_2(\Delta T)^2 + b_3\Delta W + b_4(\Delta W)^2 + b_5\Delta C + b_6(\Delta C)^2 + b_7\Delta T\Delta W + b_8\Delta T\Delta C + b_9\Delta W\Delta C| \quad (6)$$

This functional form estimates parameters that may individually be negative but which together result in a non-negative standard deviation for any CTW value being considered. At baseline, this functional form representation has standard deviation $\sigma_{baseline} = |b_0|$ as opposed to the required $\sigma_{baseline} = 0$ in Equation (3). This is done for numerical reasons and is addressed with the prior for $b_0 \sim N(0, 0.01^2)$. This constrains the value of $b_0$ to be between -0.02% and 0.02% with 95.45% probability, reflecting that $b_0$ should be as close to 0 as possible without causing numerical solver issues. This results in $\sigma_{baseline}$ values between 0% and 0.02% and therefore $0\% \le \Delta Y_{baseline}^{emulated} \le 0.02\%$, which we judge acceptable for incorporating into GCAM as a multiplier. All other parameters have very broad priors:

$$b_0 \sim N(0, 0.01^2)$$
$$a_i, b_i \sim Uniform(-300, 300) \, \forall a_i, b_i, i \neq 0 \quad (7)$$

The functional form for $\mu_{CTW}$ is equivalent to estimating the coefficients of a third order Taylor polynomial, which can approximate a wide variety of functions fairly well. Similarly, the functional form for $\sigma_{CTW}$ is conceptually related to estimating the coefficients of a second order Taylor polynomial. Because of the C3MP experimental design, emulating yield changes throughout the 21st century using Equations (1)-(7) does not require extending beyond the range of mean growing season CTW values used to train the Persephone V1.0 response functions. These functional forms are an evolution from C3MP's hybrid polynomial (Ruane et al., 2014). An exploration of other functional forms to address potential overfitting is included in Appendix A. Ruane et al. (2014) also reviews previous emulator forms across the literature, including discussion of the potential to look at non-linear terms such as killing degree days used in Schlenker and Roberts (2009), for example.

From the model defined by Equations (1)-(7), we construct the three Persephone v1.0 response functions for each production group:

Mean response: $\Delta Y_{CTW}^{emulated} = \mu_{CTW}$; $\Delta Y_{baseline}^{emulated} = \mu_{baseline} = 0\%$

High response: $\Delta Y_{CTW}^{emulated} = \mu_{CTW} + |\sigma_{CTW}|$; $\Delta Y_{baseline}^{emulated} \in (-0.02\%, 0.02\%)$ with 95.45% probability $\quad (8)$

Low response: $\Delta Y_{CTW}^{emulated} = \mu_{CTW} - |\sigma_{CTW}| \Delta Y_{baseline}^{emulated} \in (-0.02\%, 0.02\%)$ with 95.45% probability

The default high and low responses are at one standard deviation of the production group yield responses (as opposed to two or three) because we are interested in scenarios that capture a range of the simulated site responses, but not the most extreme simulated site response. This does not affect how $\mu$ and $\sigma$ are fit in Persephone v1.0, only how they are used. The Persephone v1.0 code is written flexibly enough that a user more interested in capturing the most extreme simulated site response could certainly add a multiplicative factor (e.g. $\mu + 2|\sigma|$) when using $\mu$ and $\sigma$ without having to spend the computational time refitting.

## 3   Evaluation

We primarily present figures and analysis using the model and response functions defined by Equations (1)-(8) because we found these functional forms to be the most broadly optimal of those considered. To investigate overfitting, we also examined nine other possible functional form combinations of $\mu_{CTW}$ and $\sigma_{CTW}$ for each production group, defined in Equations (A1)-
(A7). Details of the cross-validation experiments used as a method of functional form selection are in the Appendix. Briefly, because we are interested in the ability of any given response function to accurately predict yield changes in response to CTW values *not* used for training, we perform leave-one (CTW test)-out cross-validation experiments for each production group. The best performing functional form at the cross-validation experiments is then the selected functional form. This can be done to find the most broadly optimal functional form (using the same functional form for all production groups, Figure A1) or to
find the best functional form for each production group (if a user wishes to vary the functional form for each production group, Table A10). This choice does not introduce additional fitting, or computational time. It is changed only by the calls to each function in the Persephone R package by the user.

Here, we quantitatively evaluate the performance of the Persephone V1.0 response functions (Equation (8)) trained on the full span of CTW values that the 99 tests represent for each production group (Section 3.1). We also present heuristic evaluations
of mean response function performance (Section 3.2).

Files with the point estimate, as well as the standard deviation of the posterior distribution, for each coefficient in $\mu$ and $\sigma$ for all 10 functional form combinations for all production groups are available (archived at https://doi.org/10.5281/zenodo.1414423) and as part of the Persephone v1.0 R package (https://github.com/JGCRI/persephone).

### 3.1   Quantitative

We categorize the performance of the Persephone V1.0 response functions trained on the full span of CTW values (mean, high, and low response, Equation (8)) for each production group based on comparing the 99 emulated yields output from the response functions to the 99 corresponding values from the C3MP simulation data: the in sample measurement of error. These are the actual response functions an end user would have and it is important to have a performance measure for them, although this is not the performance measure used to select functional forms.

The categorization is based on the normalized root mean square error (NRMS) and the comparison for each response function is as follows:

- The 99 emulated yields returned by the mean response function are compared to the mean yield response across the production group C3MP sites for each of the 99 sensitivity tests (what we call the simulated mean yields).

- The 99 emulated yields returned by the high response function are compared to the $84.135^{th}$ percentile of yield responses
across C3MP sites for each of the 99 sensitivity tests (what we call the simulated high yields). This corresponds to matching C3MP site responses at the mean plus one standard deviation level for each of the 99 sensitivity tests when the production group C3MP site responses were normally distributed for each sensitivity test.

– The 99 emulated yields returned by the low response function are compared to the $15.865^{th}$ percentile of yield responses across C3MP sites for each of the 99 senstivity tests (what we call the simulated low yields). This corresponds to matching C3MP site responses at the mean minus one standard deviation level for each of the 99 sensitivity tests when the production group C3MP site responses were normally distributed for each sensitivity test.

As noted in Willmott (1984); Legates and McCabe (1999); Snyder et al. (2017), NRMS $< 1$ is one benchmark for adequate model performance, NRMS$< 0.5$ is a benchmark for good model performance, and NRMS = RMSE = 0 is perfect model performance. We further subdivide these categories and define excellent in-sample performance as NRMS$\leq 0.25$ for all three response functions; good performance to be $0.25 < NRMS \leq 0.5$ for at least one response function and NRMS$\leq 0.25$ for at least one response function; adequate performance to be all three response functions having $NRMS < 1$ but at least one

response function with $0.5 < NRMS < 1$; and finally poor performance occurs when any one of the three response functions has $NRMS \geq 1$.

The mean response function performs excellently for all of our production groups, although the performance of the high and low response functions differs. These measures are presented in Table 1 for the response functions defined using cubic $\mu_{CTW}$ (Equation (4)) and quadratic $\sigma_{CTW}$ (Equation (6)) for all production groups. The excellent performance of the mean

response function holds across all functional form combinations explored (Table A1-A9). In the event that a user is only concerned with a mean response scenario, a shared functional form for all production groups is acceptable. A user interested in the high and low response functions may wish to use the production group specific functional form combinations listed in Table A10, which includes the in-sample performance metric for the optimal functional form for each production group. The majority of production groups (17/25) feature excellent in-sample performance while the remaining 8 production groups

feature good overall performance. For more detail than the summary tables presented here, files of results for the leave-one-out cross validation exercises for all functional form combinations for all production groups are available in the paper analysis archive.

We also present a dashboard of quantitative evaluation plots for four of our 25 production groups in Figures 5 and 4 to provide a visual interpretation of the four in-sample performance categories. Each dashboard is organized to address the

following questions:

– Top Left: For a given group, do the three representative responses span the range of sites? In this plot, individual site yield changes for each test (blue dots), are overlaid with the emulated mean, high, and low response functions evaluated for each test (black dots). Each horizontal line of points represents one of the 99 CTW sensitivity tests.

– Top Right: For a given group, how does the emulated mean for each of the 99 tests compare to the simulated mean for

each test?

– Bottom Left: For a given group, how does the emulated high response for each of the 99 tests compare to the simulated high yield for each test?

**Table 1.** Persephone v1.0 response function performance for all production groups, for cubic $\mu_{CTW}$ (Equation (4)), quadratic $\sigma_{CTW}$ (Equation (6))

| Production group[1] | Num. C3MP sites | NRMS mean[2] | NRMS high | NRMS low | In-sample Performance |
|---|---|---|---|---|---|
| c4 IRR mid | 47 | 0.010 | 0.148 | 0.112 | Excellent |
| Maize IRR mid | 45 | 0.010 | 0.164 | 0.116 | Excellent |
| Rice RFD mid | 4 | 0.044 | 0.150 | 0.195 | Excellent |
| Rice RFD tropic | 41 | 0.020 | 0.199 | 0.146 | Excellent |
| Soybeans IRR mid | 32 | 0.017 | 0.230 | 0.176 | Excellent |
| Soybeans IRR tropic | 2 | 0.039 | 0.150 | 0.170 | Excellent |
| Soybeans RFD mid | 35 | 0.016 | 0.151 | 0.145 | Excellent |
| Soybeans RFD tropic | 9 | 0.043 | 0.198 | 0.160 | Excellent |
| c3 RFD mid | 165 | 0.010 | 0.316 | 0.270 | Good |
| c4 RFD mid | 74 | 0.016 | 0.319 | 0.241 | Good |
| c4 RFD tropic | 25 | 0.019 | 0.365 | 0.177 | Good |
| Maize IRR tropic | 7 | 0.012 | 0.345 | 0.118 | Good |
| Maize RFD mid | 66 | 0.018 | 0.293 | 0.230 | Good |
| Maize RFD tropic | 20 | 0.022 | 0.407 | 0.170 | Good |
| Rice IRR tropic | 53 | 0.088 | 0.339 | 0.261 | Good |
| Wheat IRR mid | 61 | 0.024 | 0.372 | 0.380 | Good |
| Wheat IRR tropic | 8 | 0.076 | 0.382 | 0.329 | Good |
| Wheat RFD mid | 103 | 0.021 | 0.302 | 0.280 | Good |
| Wheat RFD tropic | 4 | 0.093 | 0.364 | 0.311 | Good |
| c3 RFD tropic | 63 | 0.024 | 0.757 | 0.546 | Adequate |
| c4 IRR tropic | 14 | 0.012 | 0.998 | 0.214 | Adequate |
| Rice IRR mid | 6 | 0.029 | 0.656 | 0.427 | Adequate |
| c3 IRR mid | 103 | 0.012 | 1.038 | 0.701 | Poor |
| c3 IRR tropic | 67 | 0.072 | 1.662 | 0.790 | Poor |
| Sugarcane RFD tropic | 12 | 0.047 | 1.382 | 1.162 | Poor |

1. "IRR" = irrigated, "RFD" = rainfed, "mid" = mid-latitudes (30- 70°S, 30-70°N), "tropic" = 30°S to 30°N. 2. Note that the mean response function performs "excellent" for all production groups.

- Bottom Right: For a given group, how does the emulated low response for each of the 99 tests compare to the simulated low yield for each test?

Figure 4 displays one performance dashboard from each in-sample performance category for the broadly optimal, shared functional form cubic $\mu_{CTW}$ and quadratic $\sigma_{CTW}$ (Equations (4)-(6)), to aid interpretation of Table 1 (and Tables A1-A9).

As indicated in Table A10, any production group can be fit to result in response functions with an in-sample performance of good or excellent, if a user is willing to vary the functional forms used for each production group. Figure 5, left, presents the dashboard for one of the production groups that featured poor performance when the common functional form cubic $\mu_{CTW}$ and quadratic $\sigma_{CTW}$ (Equations (4)-(6)) was used for all production groups: rainfed sugarcane in the extended tropics. Figure 5, right, presents the dashboard when the response functions are based on the production group specific functional forms selected by cross-validation (Table A10): C3MP $\mu_{CTW}$ (Equation (A2)) and cubic $\sigma_{CTW}$ (Equation (A7)). The high and low response functions perform better in the latter case, though it is at the cost of a slightly worse (but still excellent) mean response function performance. Examination of the sugarcane entry in Tables 1, A1-A9 indicates that a cubic description of $\sigma_{CTW}$ (Equation (A7)) leads to better high and low response function performance than a quadratic representation (Equation (A6)), regardless of functional form used for $\mu_{CTW}$ (Equations (A1)-(A5)). In other words, the uncertainty across C3MP site responses for each CTW test requires a more detailed Taylor series approximation to describe. This is also generally the case for the other production groups that rated adequate or poor in-sample performance in Table 1: sometimes the C3MP individual site yield responses are distributed in such a way for each CTW test that a more flexible fit for $\sigma_{CTW}$ is necessary. Perhaps unsurprisingly, this usually occurs for either very broad production groups (such as those based on C3-photosynthesis), or for production groups with very few C3MP site outputs (irrigated rice in the mid-latitudes) rather than due to a discernible biophysical trend or requirement.

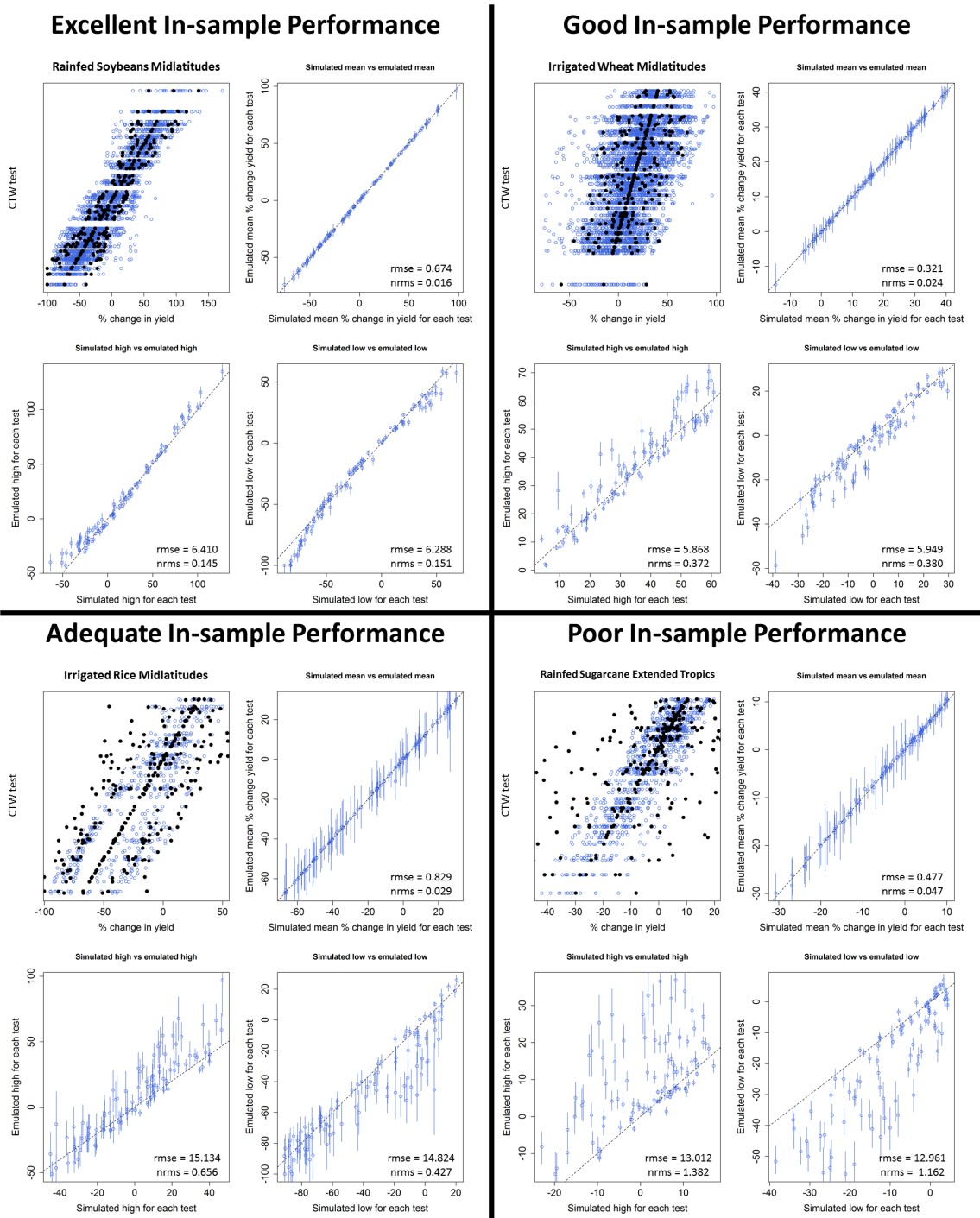

**Figure 4.** Top left: Rainfed soybeans in the mid-latitudes, an example of the excellent in-sample performance category. Top right: Irrigated wheat in the mid-latitudes, an example of the good in-sample performance category. Bottom left: Irrigated rice in the mid-latitudes, an example of the adequate in-sample performance category. Bottom right: Rainfed sugarcane in the extended tropics, an example of the poor in-sample performance category (also seen in Figure 5, left). Vertical error bars indicate 95% credible interval for each of mean, high, low emulated responses.

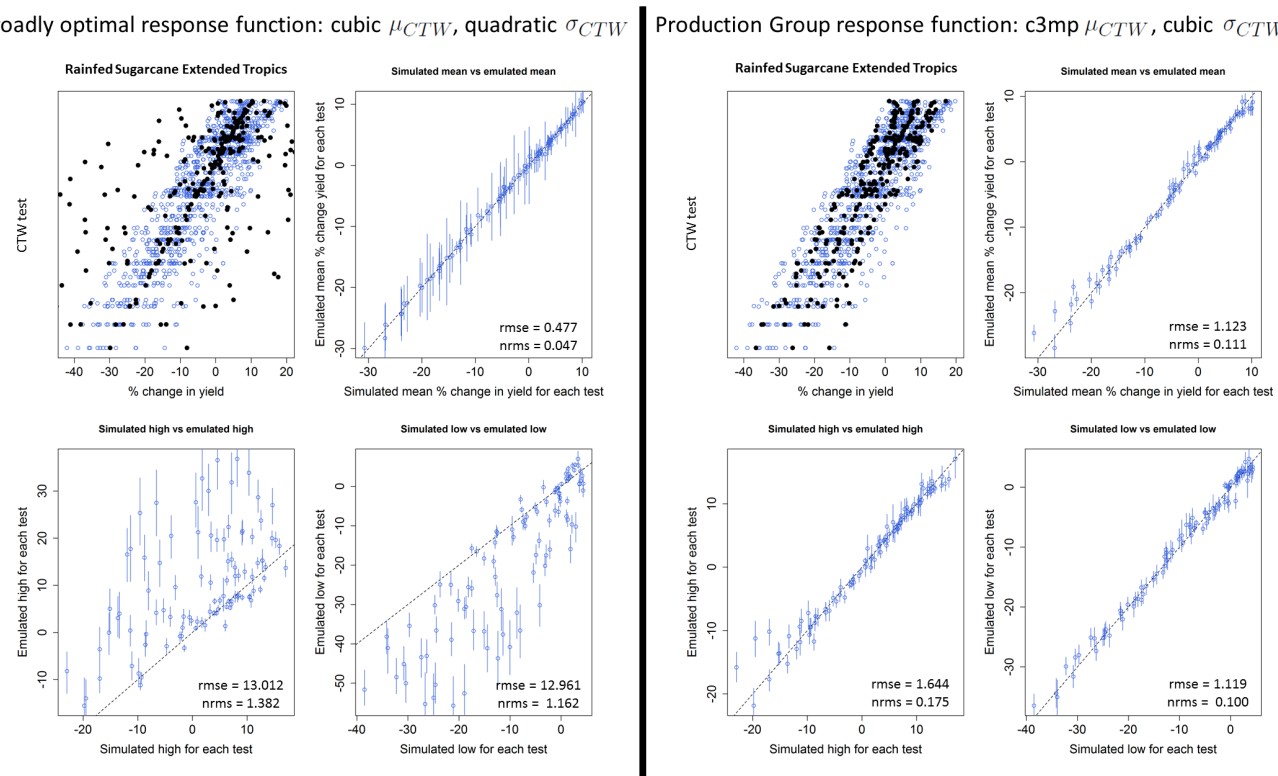

**Figure 5.** Rainfed sugarcane in the extended tropics. Left: The performance dashboard for the most broadly optimal functional form representations (i.e. if we want to use the same functional form combination for all production groups), and for which the high and low response functions poorly reproduce the simulated high and low yields for each of the 99 tests. Right: The performance dashboard for the production group specific functional forms (i.e. if we want the functional form to vary by production group). Vertical error bars indicate 95% credible interval for each of mean, high, low emulated responses.

### 3.1.1 Production groups with small sample size

It is worth noting that 7 of the 25 production groups considered here are characterized by fewer than 10 C3MP sites (Table 1). For all of these groups, it is possible to fit high and low response functions that capture the spread of the group's C3MP site responses well (Figures 6 and 7). For many of these groups, the spread in response is relatively small. The Persephone framework does not fail, rather the data upon which the V1.0 response functions are trained is imperfect and would be improved by greater density in spatial sampling. Had the spatial disaggregation used in forming production groups resulted in small sample size groups with more significant spread in site response, the Persephone framework is unlikely to represent the full spread of the sample. As this is not the case, it is left to an eventual user to judge whether such responses serve their purpose.

Figure 6 highlights this fact for the production group with smallest sample size, irrigated soybeans in the Extended tropics. The spread of C3MP sites as well as the performance dashboards for the shared optimal functional form (as from Table 1) and for the group-specific optimal functional form (Table A10). While the shared optimal functional form (middle panel)

overestimates the small spread between the two C3MP sites, the group-specific optimal functional form (right panel) captures the spread well.

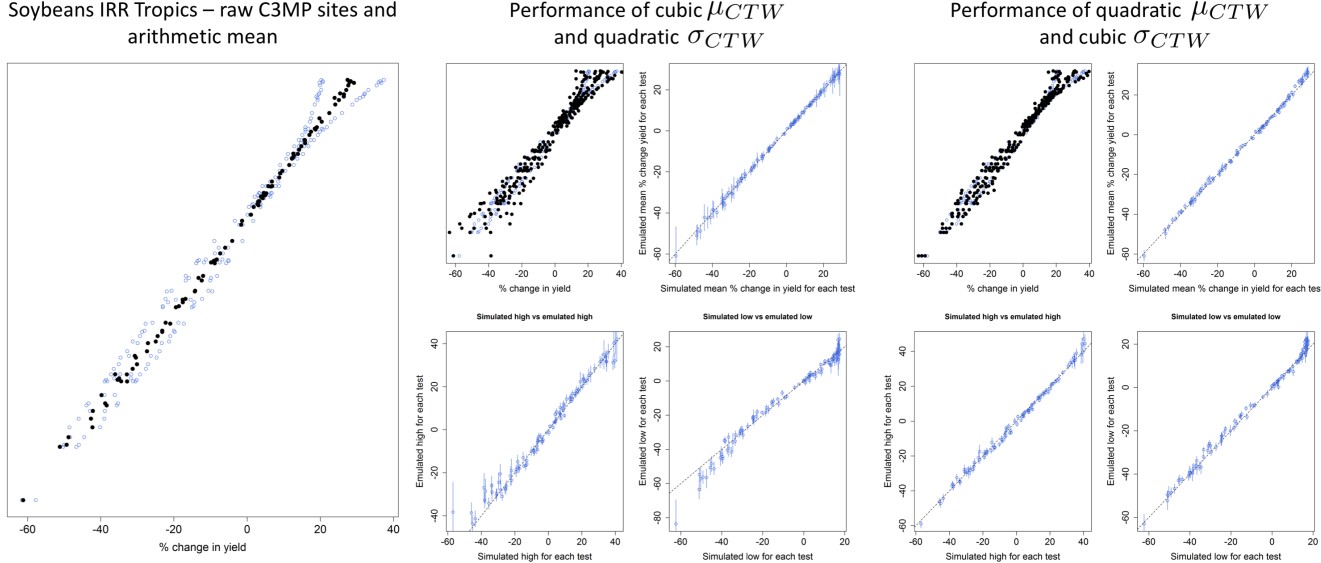

**Figure 6.** Left:The spread of yield responses for the two C3MP sites making forming the irrigated soybeans in the extended tropics production group. Middle: The performance dashboard of the shared optimal functional from (Table 1) for this production group. Right: The performance dashboard of the group-specific optimal functional form (Table A10) for this production group.

Figure 7 repeats this analysis for the next three smallest sample size groups: rainfed wheat in the extended tropics (Top), rainfed rice in the mid-latitudes (Middle), and irrigated ice in the mid-latitudes (Bottom). In all three cases the group-specific optimal functional form represents the spread of the data well. This is also the case for the two remaining production groups with fewer than 10 C3MP training sites: irrigated wheat in the extended tropics and rainfed soybeans in the extended tropics (not shown).

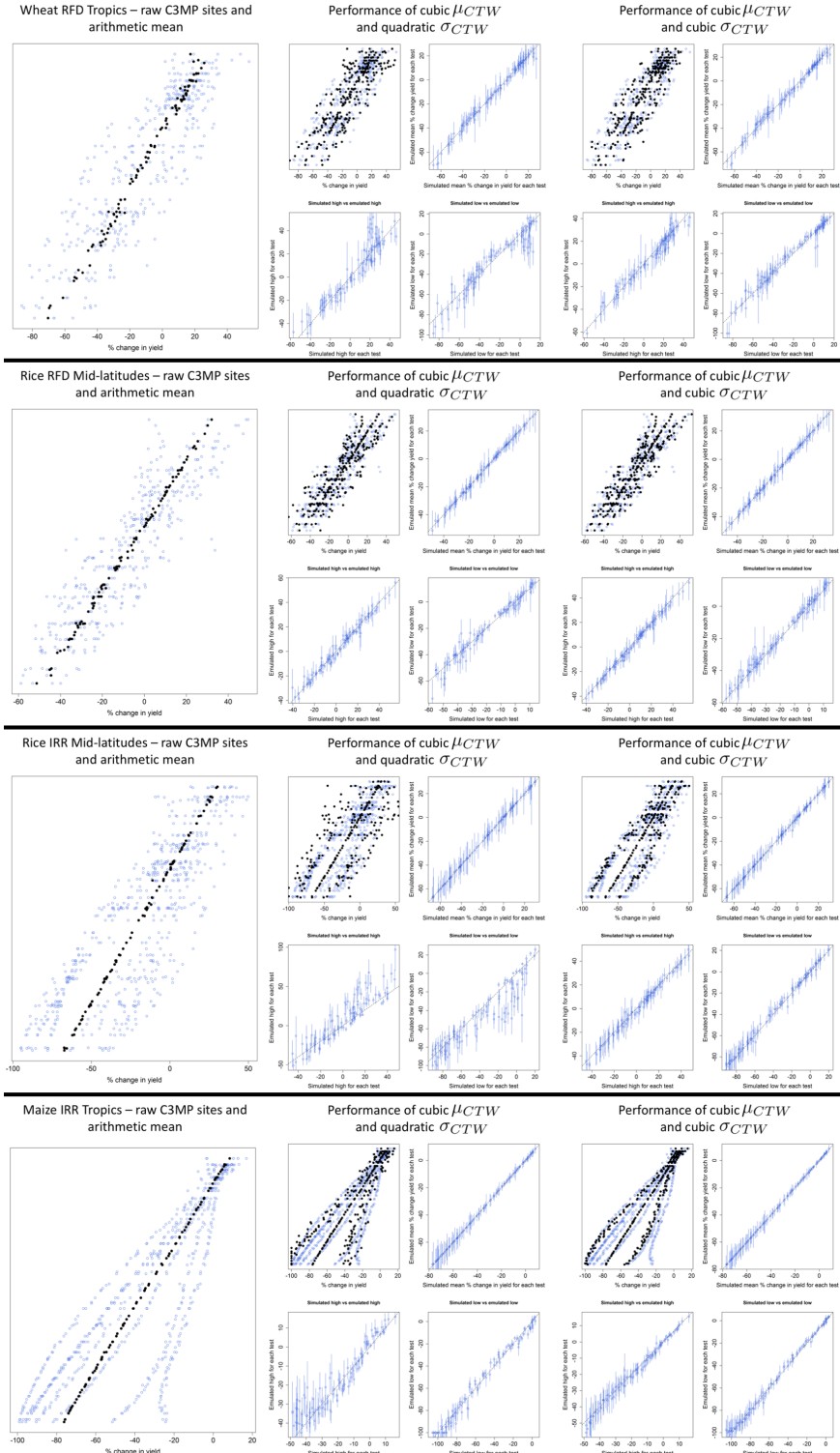

**Figure 7.** The same arrangement of figures as in Figure 6, for the rainfed wheat in the extended tropics (Top row), rainfed rice in the mid-latitudes (Second row), irrigated rice in the mid-latitudes (Third row), and irrigated maize in the extended tropics (bottom row) production groups.

## 3.2 Qualitative

One motivation for the 25 production groups based on [maize, wheat, rice, soybeans, C3, C4 (minus sugarcane), and sugarcane] X [irrigated or rainfed] X [extended tropics or mid-latitudes] is to evaluate emulator performance beyond the quantitative. Given that some GCAM users will only be interested in the mean response functions, it is particularly important to validate that these

5 functions capture key biological features of each crop, beyond the quantitative agreement for the 99 C3MP tests measured by the in-sample performance metric in Section 3.1. In particular, these are features motivated by biophysical intuition and present in most of the C3MP sites. Therefore we verify that these features are retained in the emulator.

We use impact response surfaces to visualize these features, examples of which are given in Figures 8 and 9. The three-dimensional CTW space is most easily examined by looking at cross sections where one of the CTW dimensions is kept

constant while the other two vary. The brown to blue color bar in each of these figures depicts contours for the value of the mean yield response ($\mu_{CTW}$) while the overlaid labeled black lines depict contours representing uncertainty ($\sigma_{CTW}$, used to create the high and low response functions).

We first identify three important relationships we would expect a successful emulation of C3MP mean responses (brown to blue color bar) to obey:

– C3 crops respond strongly and positively to increases in global $CO_2$ concentrations; C4 crops have noticeably less benefit from $CO_2$ increases.

– Agriculture in the tropics tends to response more negatively/less positively to changes in temperature than agriculture in the higher latitudes as the extended tropics correspond to a higher baseline temperature.

– Irrigated crops have almost no response to changes in precipitation, whereas rainfed crops do.

These benchmarks are met: Figure 8 features impact response surfaces that highlight the C3-photosynthesis and C4-photosynthesis difference, the rainfed and irrigated difference, and the latitude difference. The full collection of impact response surfaces for all production groups are included in the paper analysis archive. These benchmarks for the mean response are met in those as well. When there are exceptions, we have investigated to find that the mean response function is faithfully representing the underlying C3MP data and that it is the sampling of C3MP sites making up the production group responsible for the

discrepancy. Note that, in Figure 8, uncertainty is greatest in the $CO_2$-precipitation and $CO_2$-temperature slices, and increases with larger changes from the baseline condition. This follows with current practices for the process-based crop models forming the C3MP data set: $CO_2$ is clearly related to yields but the details of this relationship are highly uncertain and implemented differently across process-based, site specific crop models.

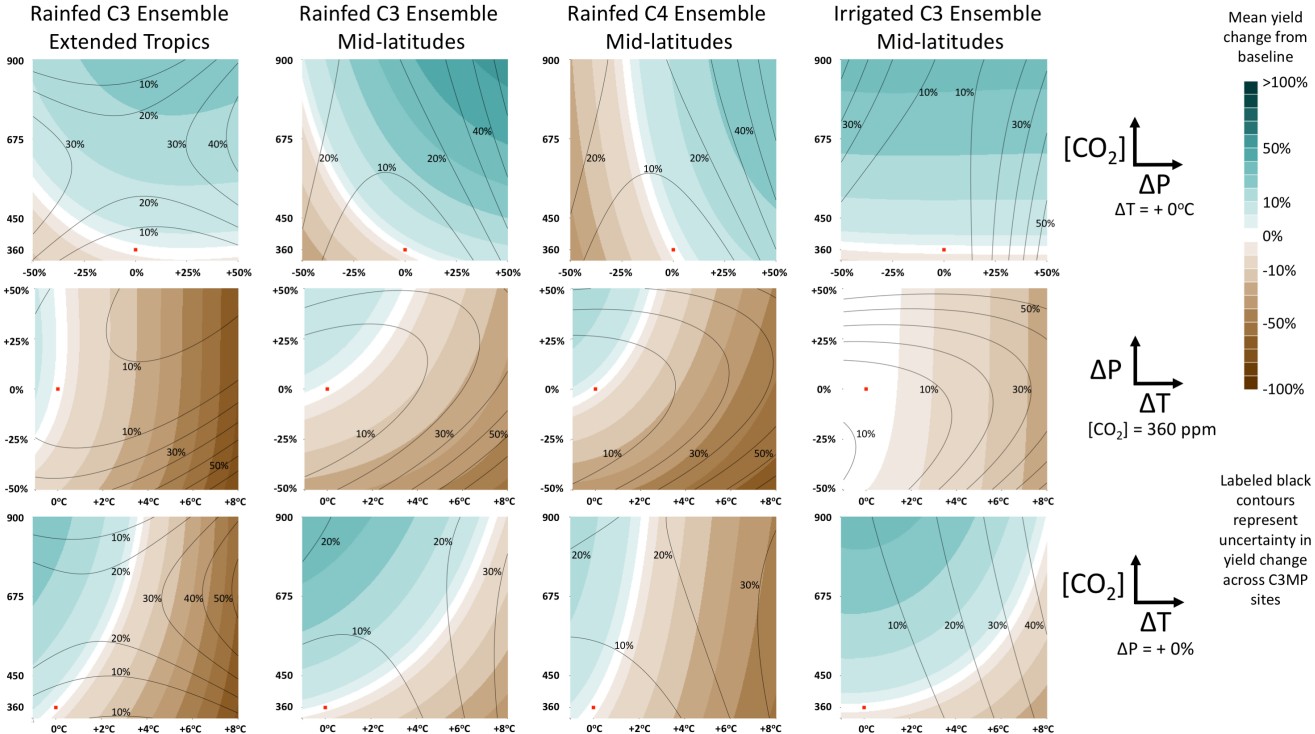

**Figure 8.** Select impact response surfaces - a collection of 2-parameter slices of our 3-parameter space (not a visualization of the full space). The color represents the yield change for a given local CTW perturbation as a % of baseline yields (1980-2009 planting year average, position shown as red square). Labeled black contours are uncertainty across the submitted site specific crop models.

The *pattern* of yield response to CTW changes appears to be more qualitatively consistent across C3MP sites than the quantitative differences across sites (for example, Figure 3). Figure 9 displays this pattern for one cross-section of CTW space for 12 of 66 rainfed maize sites in the mid-latitudes, and for the emulated mean response. While the actual numerical values of the response surfaces differ at each site, the pattern of response seen at most sites (increasing yield with high $CO_2$ and low temperature changes in the upper left, decreasing yields elsewhere) is consistent and shared by the emulated mean response. The high and low response functions are able to capture much of the quantitative spread in site responses, though, as noted in Section 2.3, not the most extreme sites. We specifically included the sites at Ames, IA, Naousa, Greece, and Lublin Poland because they feature the most qualitatively different patterns. The pattern at the 54 sites not displayed closely resemble the other 9 sites in Figure 9. This pattern is seen in the broader impact response surfaces literature (Ruane et al., 2014; Pirttioja et al., 2015; Fronzek et al., 2018) as well, further improving confidence in the emulated mean response. All individual site impact response surfaces are included in the paper analysis archive.

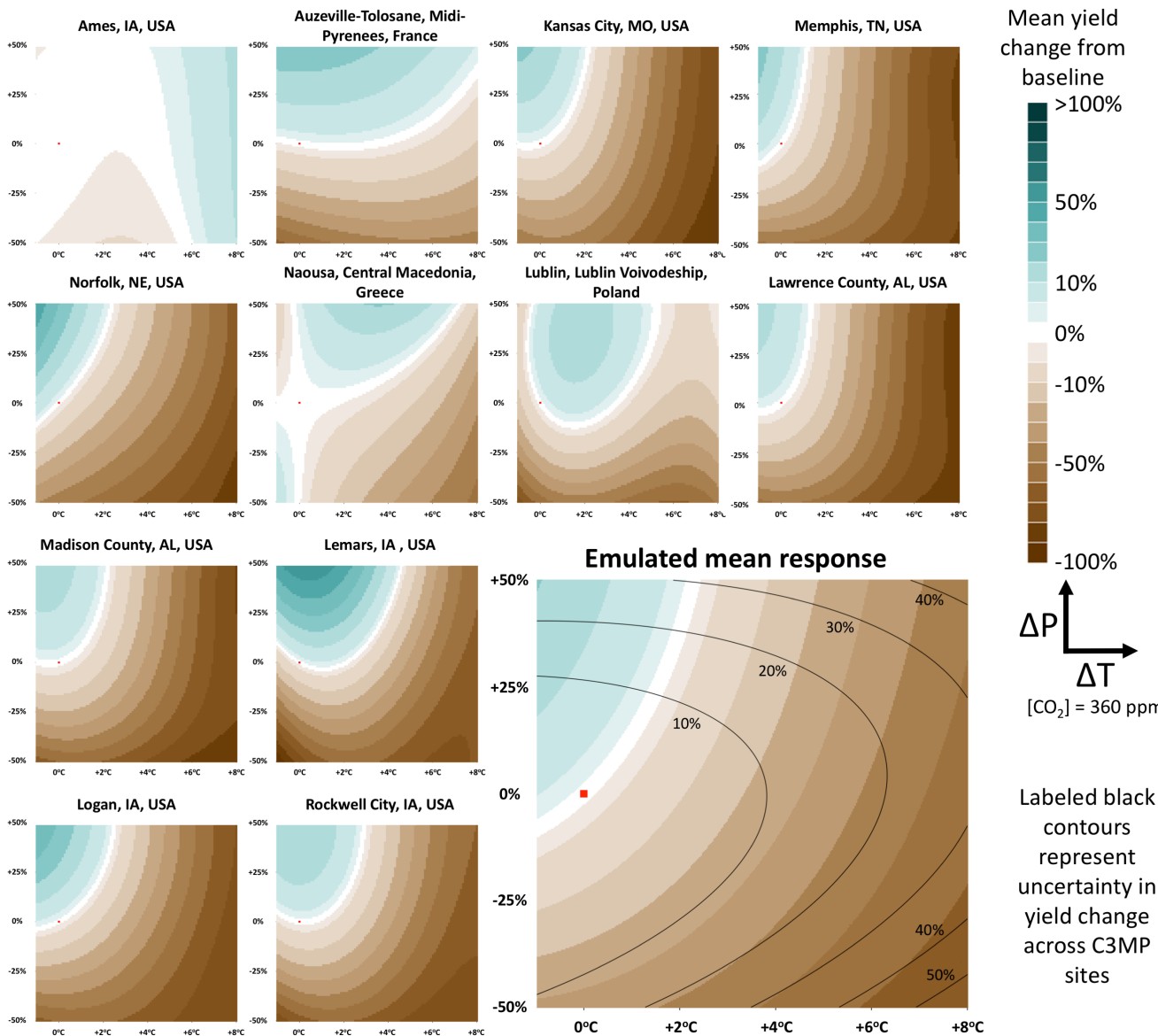

**Figure 9.** Yield responses to changes in temperature and precipitation with fixed [CO$_2$] = 360 ppm for 12 (of 66 total) rainfed maize sites located in the mid-latitudes, as well as the emulated mean response for use in GCAM.

## 4 Applications

Figure 10 demonstrates the basic procedure followed in using Persephone within GCAM (using the average of 2071-2100 HadGEM2-ES RCP 8.5 projections as an example). The first requirement is a global gridded file of local precipitation and local temperature drawn from climate projections, along with a global CO$_2$ concentration level. Temperature and precipitation

changes are calculated only for the relevant local growing season months in comparison to a 1980-2009 baseline value. The different maps of local temperature and precipitation changes on the left side of Figure 10 reflect that there are differences in the dates of the local growing season for rainfed maize and wheat. Note that this includes a global $CO_2$ concentration of 812 ppm, compared to the baseline level of 360 ppm. The $CO_2$ change alone leads to increased yields for rainfed wheat mid-

latitude even in the absence of changes in temperature and precipitation. Indeed, the higher $CO_2$ elevates yields (compared to the baseline) across all but the most extreme hot and dry conditions. Conversely, the yield response for rainfed tropical maize is barely helped by elevated $CO_2$.

In a typical RCP 8.5 scenario, there are sometimes a few grid cells with local precipitation changes that are out of sample. We convert these out of sample points to the extreme of our sample so that we avoid extrapolation (eg a 74% local increase

in precipitation gets the response of 50% increase in precipitation - the maximum response to increased precipitation). Note that many of these large percentage changes in precipitation are actually the symptoms of ESM biases or small precipitation changes in arid regions that are unlikely to have agriculture. Holding to 50% precipitation change likely improves the fidelity of these estimates (Ruane et al., 2014).

The second step in using Persephone for GCAM is that CTW changes for each grid cell with climate data are passed into

the Persephone V1.0 response functions (depending on species/management/latitude zone) to create the desired global gridded map of yield changes that would represent the likely agricultural response. The abrupt change in behavior across 30°N and 30°S (particularly noticeable for wheat in Southern Asia) are due to our division of training data into mid-latitudes and extended tropics production groups. Those abrupt changes will soften as these impacts are aggregated to the larger GCAM land region level before being applied as multipliers in the experiments outlined in Figure 1.

Figure 11 presents the rainfed maize impact response surfaces and yield change maps for the bias-corrected ISIMIP entry of HadGEM2-ES RCP 8.5 (Warszawski et al., 2014) 2071-2100 average CTW changes (displayed in Figure 10) for the low (left), mean (center) and high (right) response functions. The high and low response surfaces result from adding or subtracting the gray uncertainty contours to the brown-blue mean yield response contours in the mean response surfaces (Equation (8)). Note that under the high response function, there are a few regions that experience increased yields due to large increases in

precipitation offsetting temperature increases. The differences in these three response functions will allow the boundaries of crop response uncertainty to be run through GCAM, resulting in a spread of socioeconomic and environmental impacts in response to a particular future climate.

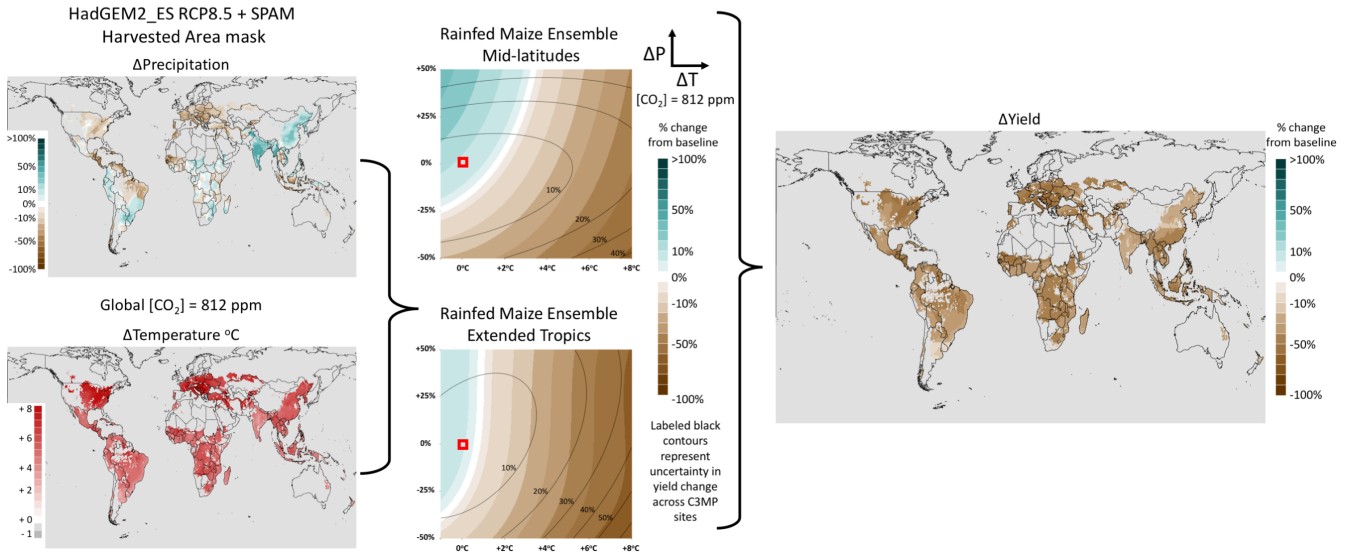

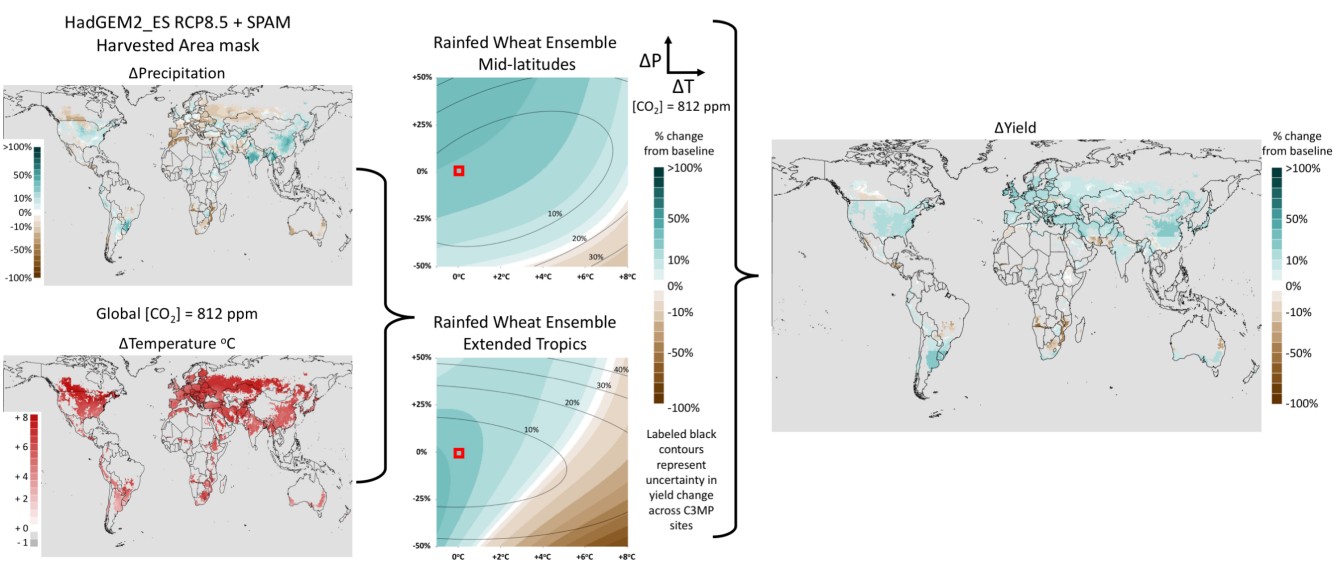

**Figure 10.** Tracing the path from gridded local growing season temperature and precipitation changes and global $CO_2$ = 812 ppm concentration under HadGEM2-ES RCP 8.5 for 2071-2100 compared to 1980-2009, through the relevant yield response functions (represented here as impact response surfaces) to generate mean yield change maps for Rainfed maize (top) and rainfed wheat (bottom). The open red square is placed at no change in temperature and precipitation for each impact response surface. For plotting clarity, we use a harvested area mask of grid cell harvested area > 10 hectares in the SPAM 2005 data set (You et al., 2014)

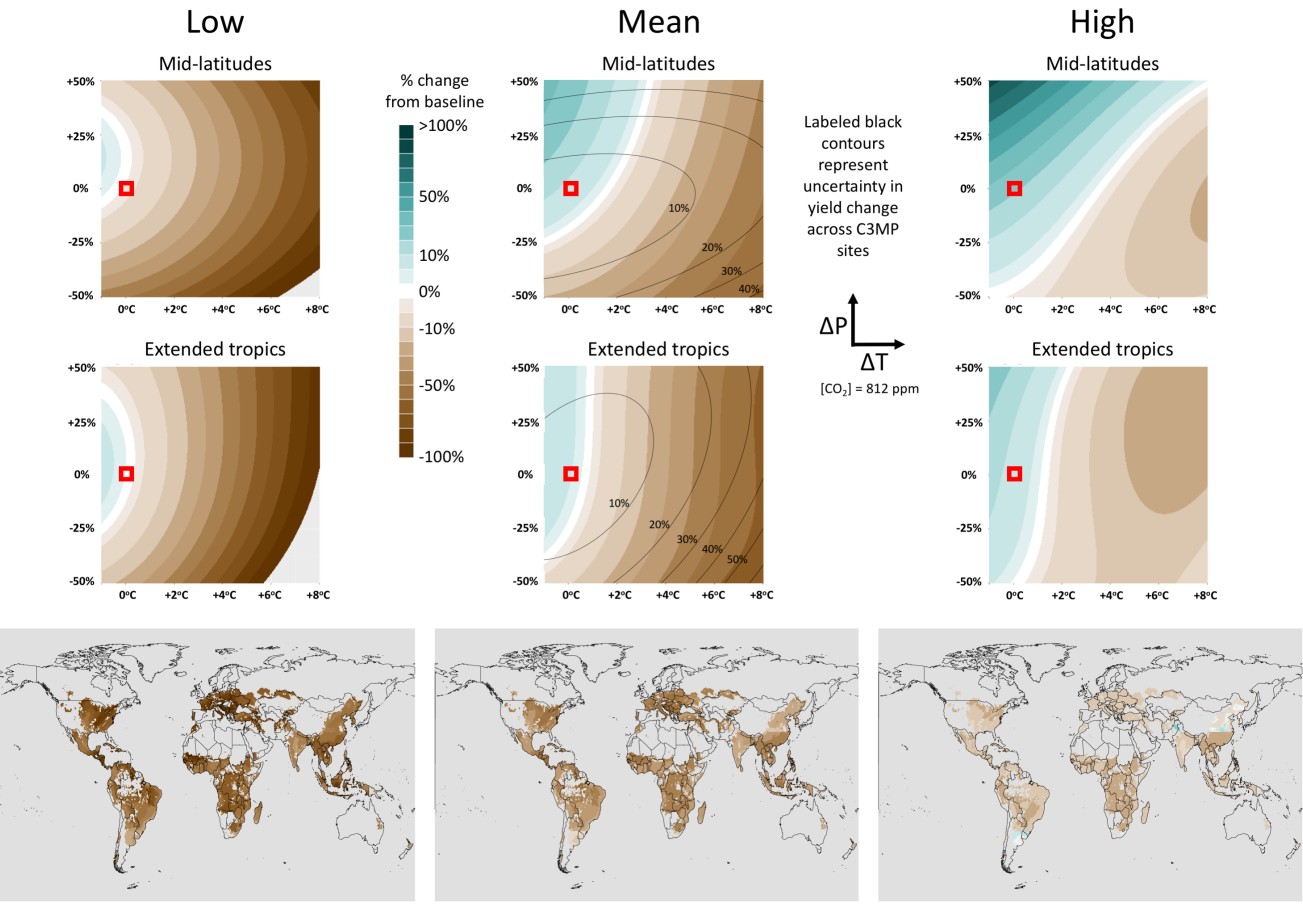

**Figure 11.** Low, mean, and high response surfaces for the mid-latitudes (top row) and extended tropics (middle row) for rainfed maize, as well as the resulting maps of yield changes under the same HadGEM2-ES RCP 8.5 2071-2100 CTW changes as Figure 10.

## 4.1 Comparison to other crop modeling results

We further examine the Persephone V1.0 response functions driven by HadGEM2-ES RCP 8.5 CTW changes by comparing our results with previous AgMIP global gridded crop model (GGCM) yield change data released under ISIMIP (Rosenzweig et al., 2014; Warszawski et al., 2014). In order to compare the best possible emulation of the C3MP data set to the range of

5 AgMIP/ISIMIP GGCM results, the production group-specific optimal functional forms provided in Table A10 are used here. To have the most direct comparison possible, the ISIMIP GGCM yield time series were converted from actual yield values to percent changes from 1980-2009 baseline yields. It is important to note that the GGCMs were driven by historical climate data from 1980-2004. 2005-2009 yield data for each GGCM was driven by HadGEM2-ES RCP 8.5, given that this was considered a "future" simulation according to the GCM projections from 2005 forward. The results from the GGCMs which include model-

10 specific [$CO_2$] effects were used. Both Persephone V1.0 and the ISIMIP GGCM yield change data are compared only on grid cells with harvested area > 10 hectares in the SPAM 2005 data set (You et al., 2014).

As the ISIMIP GGCMs did not directly participate in the C3MP exercise, no version of these GGCMs was used in the training data that produced the Persephone V1.0 response functions, and there is no *a priori* reason to expect the Persephone V1.0 range of yield changes to match the ISIMIP range. The site-specific simulations using various versions of DSSAT submitted to the C3MP exercise feature different configurations and model versions than the ISIMIP GGCM pDSSAT (a global gridded implementation of DSSAT). Given this fact, and that the C3MP archive includes results from non-DSSAT site-specific crop models, there is again no expectation of replicating pDSSAT results even though the fundamental crop responses are similar. Finally, it is also worth noting is that the 1980-2009 historical/RCP8.5 HadGEM2-ES simulation is not the same as the historical, site specific and AgMERRA data used by modelers submitting to C3MP. This combination of different responses and different baselines across C3MP and the ISIMIP GGCMs means there could be considerable differences in interannual variability and mean yields, which may be a reason that the Persephone V1.0 response functions may predict different yield changes from the ISIMIP range for some crops.

However, it is still worth evaluating our results against the GGCM data. Figure 12 compares the range of aggregated (via MIRCA2000 harvested area Portmann et al., 2010), time averaged 2071-2100 yield changes from Persephone V1.0 response functions to the range of ISIMIP yield changes at the global level (top), in the extended tropics latitude band (bottom left) and in the mid-latitudes band (bottom right) for both irrigated and rainfed maize, rice, soybeans, and wheat. For context, in the time since the AgMIP/ISIMIP results were published, the IMAGE-LEITAP model has been largely abandoned. Further, IMAGE-LEITAP, LPJ-GUESS, and LPJmL feature relatively unlimited nutrient constraints, resulting in frequent yield increases given an unconstrained $CO_2$ response. For many of the production groups, the range of Persephone V1.0 yield changes lies at least partially within the ISIMIP range, suggesting that the response functions for those production groups result in yield changes consistent with ISIMIP. Those production groups that differ substantially from the ISIMIP yield range are due to underlying differences in the C3MP data set versus those produced from the ISIMIP GGCMs. That is, while the Persephone framework emulates the C3MP data well, response functions based on a different data set may behave more consistently with the ISIMIP GGCMs given differences between the model selection and local farm system configurations of the C3MP and GGCM ensembles.

Of the production groups with yield ranges much smaller than the range of ISIMIP yield changes, several (irrigated and rainfed soybeans in the extended tropics and irrigated and rainfed rice in the mid-latitudes) are small sample size groups (Table 1, Section 3.1.1). Future coordinated sensitivity studies of site-specific crop models would ideally include more participation in a broader range of regions, but this is a current limitation of the Persephone V1.0 response functions. This adds additional support to the call for a designed network of site-based crop models, intended to cover all regions and systems, to participate in coordinated sensitivity studies raised in Ruane et al. (2017).

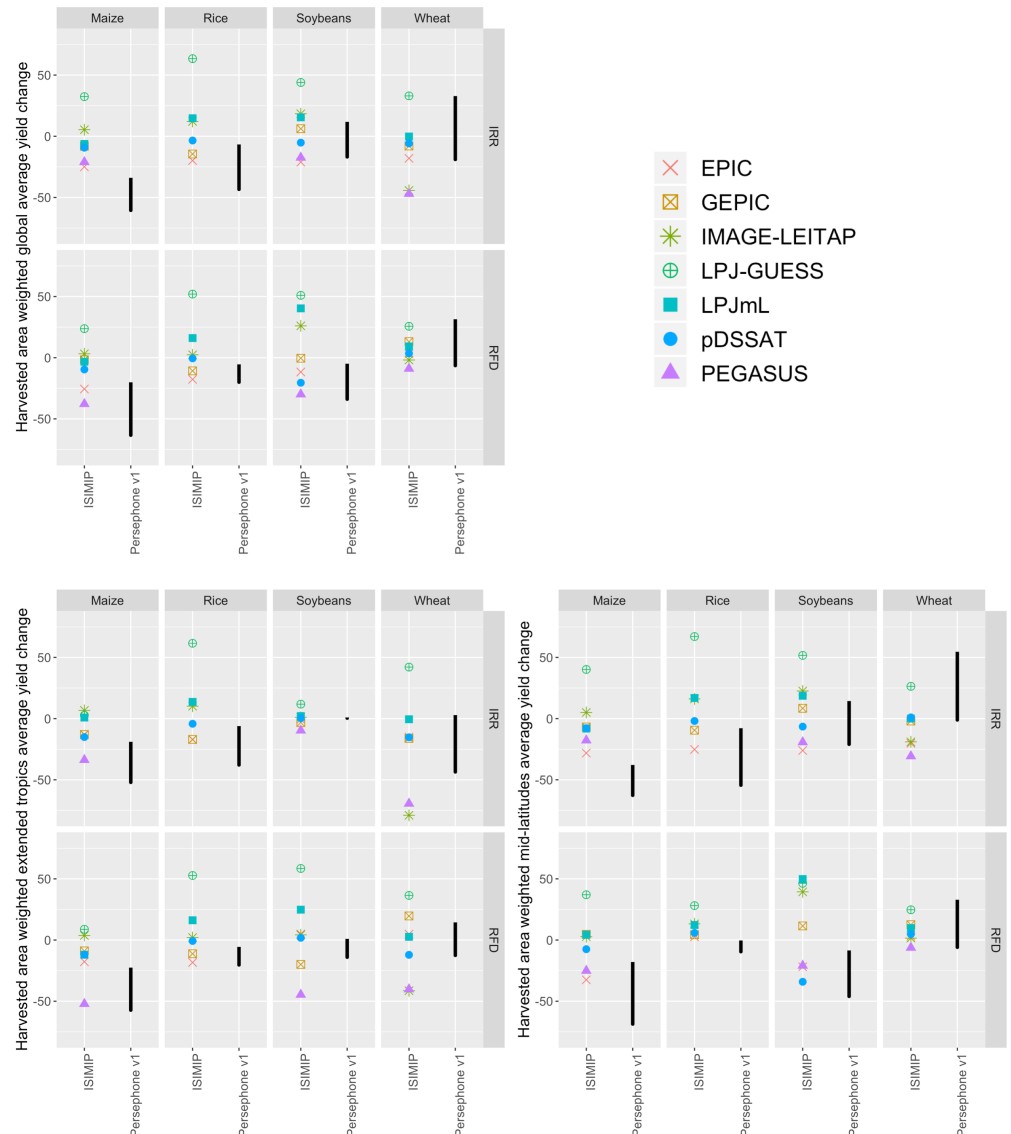

**Figure 12.** Aggregated (via MIRCA2000 harvested area (Portmann et al., 2010)), time averaged 2071-2100 yield changes for Persephone V1.0 response functions and the ISIMIP GGCM range of results for multiple production groups. Top: Comparison of global average yield changes. Bottom left: Comparison of average yield changes in the extended latitude band. Bottom right: Comparison of average yield changes in the mid-latitude band.

The Persephone V1.0 range of yield changes for irrigated maize also noticeably departs from the ISIMIP range of yield changes in the mid-latitudes (Figure 12, bottom right), which in turn drives the disagreement at the global level (Figure 12, top). This is not due to an error in emulation or due to small sample sizes (Table 1, Figure 7, bottom), but rather due to a fundamental disagreement in the predicted maize response among the site-specific crop models of C3MP and the ISIMIP

GGCMs. It is worth noting that yield changes predicted by Persephone V1.0 response functions are consistent with work examining maize site data. Namely, a 2014 site-specific model comparison study by the AgMIP-Maize team found irrigated and rainfed maize yield changes in response to a local temperature +6°C of similar values to the Persephone V1.0 range of responses (see Figure 3 of Bassu et al., 2014). The HadGEM2-ES RCP 8.5 local growing season temperature 2071-2100 change map for irrigated maize used to drive the Persephone V1.0 response functions is shown in Figure 13 and it is worth noting that many major producers of maize see temperature increases of at least +6°C. Further, recent analysis of FACE experiments and crop model results suggest that maize primarily benefits from high $[CO_2]$ during drought, indicating that models of the effects of $[CO_2]$ fertilization on irrigated maize (and rainfed maize during non-drought periods) may be overly beneficial (Durand et al., 2018). This suggests that the more pessimistic irrigated maize yield changes predicted by the C3MP sites and therefore Persephone V1.0 are more consistent with site-specific crop models and FACE experiments than they are with the ISIMIP GGCM range of results. While it would be ideal to have GGCM results from more GGCMs and more recent model versions for comparison, such results are not yet public. This discrepancy between the results of site-specific crop models and FACE experiments versus GGGMs supports the call in Leakey et al. (2012) for further investigation to understand regional and system-specific variation in $[CO_2]$ response.

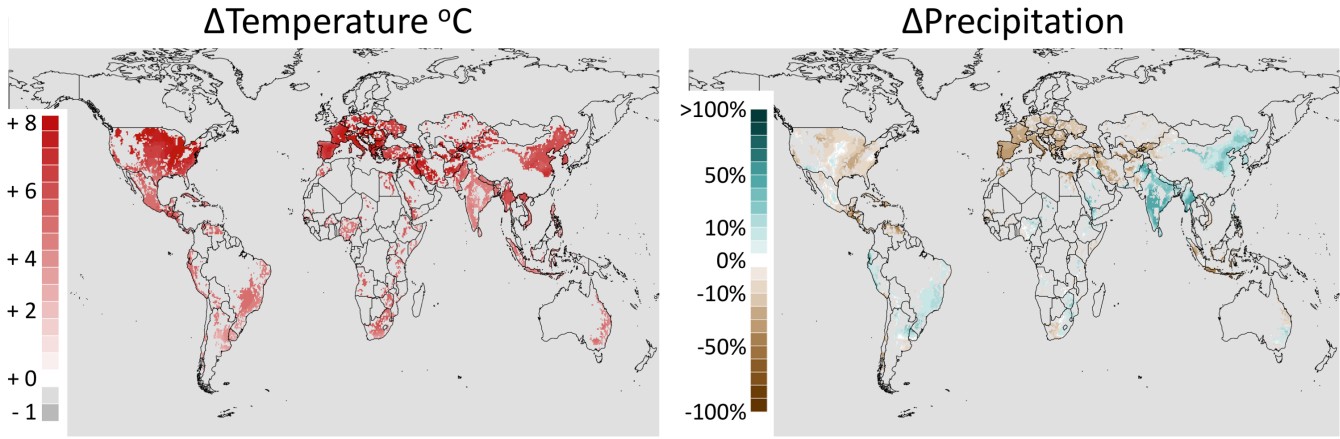

**Figure 13.** Gridded local growing season temperature and precipitation changes for irrigated maize under HadGEM2-ES RCP 8.5 for 2071-2100 compared to 1980-2009, global $CO_2$ = 812 ppm concentration.

Figure 14 includes a spatial comparison of the Persephone V1.0 low, mean, and high yield changes for irrigated maize (analogous to Figure 11) with the range of ISIMIP responses in each grid cell. Specifically, maps of the minimum, median, and maximum irrigated maize yield change across the ISIMIP GGCMs are plotted in each grid cell; no individual GGCM would produce any of these maps. As noted above, the Persephone range of yield changes in each grid cell is generally more pessimistic than the ISIMIP range, but there does appear to be spatial consistency in terms of response strength in several regions between the Persephone V1.0 range and the ISIMIP range. C3MP, and therefore the Persephone V1.0 projections, capture a strong temperature dependence and a lesser response to precipitation (particularly for irrigated crops). Because

warmer temperatures are nearly universal in the HadGEM2-ES RCP 8.5 projection (Figure 13), there is limited irrigated crop response to precipitation changes, and [$CO_2$] response for maize is small among the mechanistic models that are more prominent in C3MP than in the GGCMs, there is nothing but yield decreases in the Persephone projection. In the ISIMIP range of GGCMs, there are models that are more positively responsive to precipitation and [$CO_2$] in the C4-photosynthesis maize

5 crop, so wetter conditions and/or higher [$CO_2$] are much more beneficial in the ISIMIP maximum map (Rosenzweig et al., 2014).

## Maize Irrigated

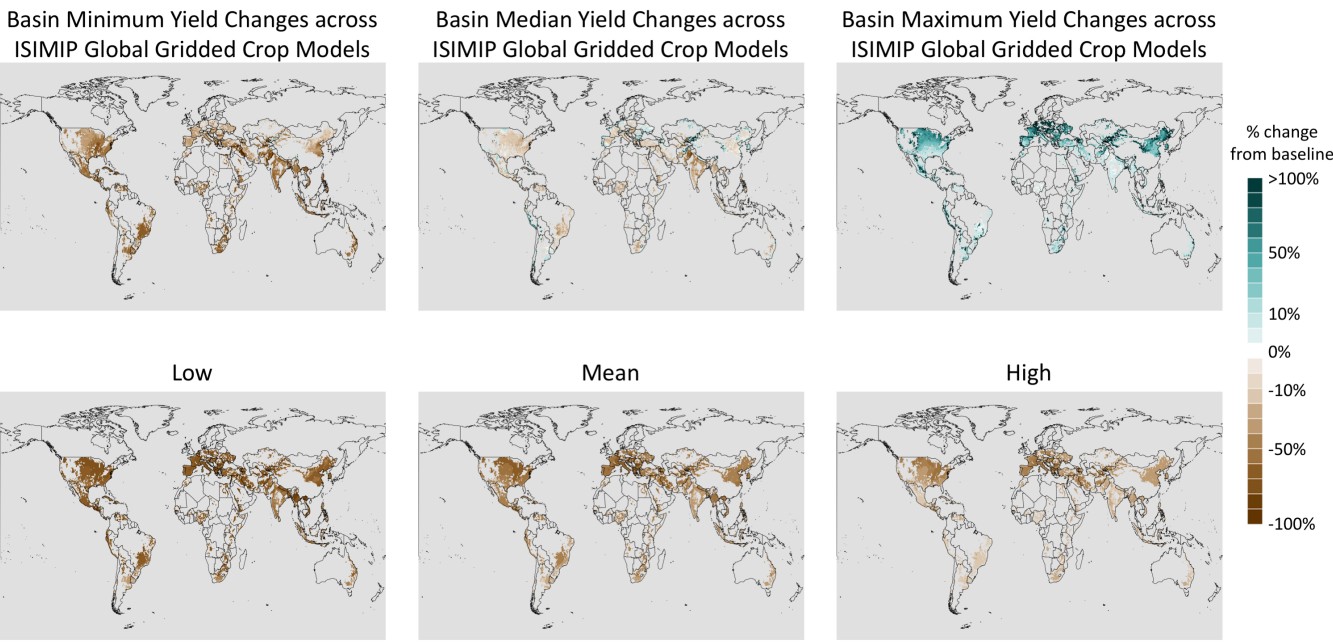

**Figure 14.** Top: grid cell specific minimum (Left), median (Middle), and maximum (Right) yield change across the ISIMIP GGCMs for irrigated maize. Bottom: the low, mean, and high Persephone V1.0 yield changes in each grid cell for irrigated maize.

Figure 15 presents the same analysis for a production group that Persephone V1.0 matches the ISIMIP global average range well: rainfed wheat. For reference, the HadGEM2-ES RCP 8.5 local growing season temperature and precipitation projections for rainfed wheat are included in Figure 10, bottom. Again, there is noticeable spatial consistency in response strength between

10 the Persephone V1.0 range and the ISIMIP range. For wheat, the C3MP models and therefore the Persephone V1.0 projections are in closer agreement with the ISIMIP GGCMs on C3-photosynthesis [$CO_2$] response and water limitations in many regions. Additionally, the harvested area mask used for rainfed wheat include many more regions that are limited by cool baseline temperatures and thus stand to gain from warmer conditions than the regions considered for irrigated maize. Put together, these observations indicate that both Persephone V1.0 and ISIMIP are capable of the large gains in the optimistic maximum model

response scenario. Together with Figure 14, this suggests that the Persephone V1.0 response functions are spatially consistent with the ISIMIP range of yield changes even when the global average ranges may disagree.

# Wheat Rainfed

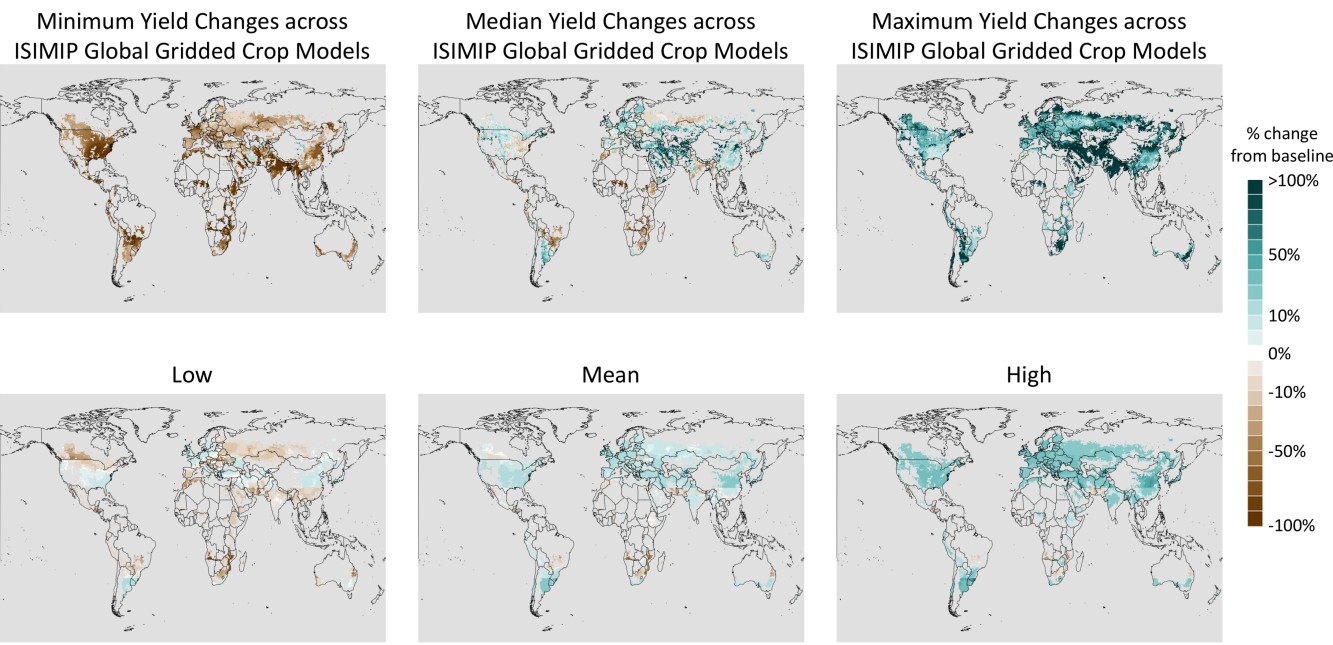

**Figure 15.** Top: grid cell specific minimum (Left), median (Middle), and maximum (Right) yield change across the ISIMIP GGCMs for rainfed wheat. Bottom: the low, mean, and high Persephone V1.0 yield changes in each grid cell for rainfed wheat.

## 5 Conclusions and discussion

We have presented the Persephone emulator framework that results in three V1.0 response functions to emulate a range of crop yield changes in response to future CTW changes for 25 production groups. The response functions are inexpensive to evaluate, open doors to new feedback loops between society and the natural environment (Figure 1), and represent multiple models and farming systems. The Persephone V1.0 response functions agree well with the underlying C3MP training data and are rapid to evaluate, with in-sample performance metrics being particularly strong for the mean response in each production group. The rapid evaluation time of the response functions, relative to a global gridded crop model, is extremely imporant given that models such as GCAM are designed to be run rapidly to trace the impacts of future scenarios (at most hours per scenario). The GCAM model development team prioritizes staying on this order of computation time, even for the planned experiments outlined in Section 2.1, because it results in a nimble, flexible model that allows multiple iterations for probability, uncertainty, and process understanding. In addition to the good quantitative agreement of our response functions with all C3MP crop-irrigation-latitude ensembles, we further evaluated our mean response function heuristically, finding that the mean responds to changes in CTW

as one would expect for comparisons across C3/C4 photosynthesis mechanisms, rainfed versus irrigated management, and latitude zones. Finally, the range of V1.0 yield changes were evaluated against a variety of past global gridded crop modeling, site specific crop modeling, and/or empirical studies for many of the production groups and found to be consistent.

As a result of the culling methods outlined in Section 2.2, 575 C3MP sites are used for training the Persephone functions. These sites account for many major crops where they are typically grown, as well as a wider variety of crops than has been examined in past studies. One key observation is that, if one were only concerned with capturing the mean response, any of the functional forms examined for $\mu_{CTW}$ (Equations (A1) - (A5)) in the Appendix would be excellent, with all five forms featuring in-sample NRMS $< 0.2$ for all production groups (Table 1). The challenge is in defining a pair of response functions, $\mu$ and $\sigma$, to characterize a range of uncertainty across C3MP site responses to each CTW change. It should also be noted that such a range of uncertainty will capture only a portion of the uncertainty in response in national and multi-national GCAM units. The Persephone framework may be used with future more spatially dense data sets to characterize this uncertainty more fully.

The modeling choices made in this study introduce a variety of caveats. Foremost, it is likely that future versions of Persephone response functions, trained on different data sets, will almost certainly result in different response functions. Yet this work has shown that the Persephone framework is well-suited to this kind of problem, and that the V1.0 response functions developed from the C3MP data emulate that data well. They also perform reasonably well on heuristic metrics and in comparison to other crop modeling efforts. Another important caveat is that GCAM operates on five year timesteps. Therefore, the response functions in this work only characterize yield responses to long-term, local Earth system state changes. Capturing interannual variability and responses to abrupt weather shocks is an area that may form future phases of this research. We note that this is a more difficult task, given that year-to-year variability depends on many factors that tend to average out over longer terms (e.g. intra-seasonal variability such as heat waves or dry spells). Additionally, this work did not account for differing nitrogen application rates across different C3MP sites. Nitrogen data is included in the C3MP archive, but the sites are heavily biased to high nitrogen application (this is likely a function of the most commonly simulated sites also being systems with higher input investment). There are also a number of sites with no recorded nitrogen information, which were kept for this study. With so few sites featuring low nitrogen application rates, we considered examining the nitrogen dimension of yield responses to be its own intellectual challenge reserved for future work, the methods of which will likely be determined by the desired use. Similarly, exploration of forming production groups based on different crop groups, different latitudinal zones, Koppen-Geiger or temperature zones would require trivial changes, limited only by the number of sites available to sort into different production groups. Finally, it is worth noting that any emulator is only as good as the data upon which it is trained. If crop modeling studies that provide data to an emulator do not account for real-world behaviors, the emulator will not capture such behaviors either.

For clear analysis in this paper, we have presented results for the functional form combination that performed best at the cross-validation experiments described in the Appendix for the most production groups. Therefore one remedy to the presence of ensembles with poorer emulator performance on in-sample metrics (Table 1) would be to use different functional forms for each production group to create a more globally optimal set of response functions. These are laid out for each production group in Table A10, along with the in-sample performance of the group-specific optimal functional forms. Some analysis with

these production group-specific optimal models is included in Section 3.1.1 and Section 4.1. The data processing, emulator fitting, and analysis techniques presented in this paper are agnostic of the actual functional forms used for $\mu_{CTW}$ and $\sigma_{CTW}$ as long as they are linear-in-parameters. Varying functional form by production group will only require different inputs to the Persephone R functions, not refitting of any parameters.

The most immediate future work involving Persephone v1.0 will be to fully implement the feedback loop sketched in Figure 1. Specifically, using GCAM to examine the broad impacts of a sustained drought, hypothetical or emergent from the feedback loop sketched in Figure 1, would be an excellent application of this yield change emulator. Once the illustrated links have been implemented and full runs of the loop have been timed, future development may take place. In addition to the exploration of the nitrogen dimension of yield response and allowing response functional form to differ by production group, Persephone version

2 may incorporate other predictors as data is available, explore more dynamic feature selection algorithms for functional form selection for $\mu_{CTW}$ and $\sigma_{CTW}$ such as L1-regularization (which favors sparse models), and/or be trained with data sets that may be released in the future. Which of these is explored next will depend on the outcomes of the initial full feedback loop studies with GCAM. This study represents the first vital, necessary step in better identifying a pathway in which society can develop with balanced consideration of the natural environment and managed environments like agriculture through connecting

Persephone and GCAM.

*Code and data availability.* Software implementing this technique is available as an R package released under the GNU General Public License. Full source can be found in the project's GitHub repository (https://github.com/JGCRI/persephone and https://doi.org/10.5281/zenodo.1415487). Release version 1.0.0 of the package was used for all of the work in this paper.

The data and analysis code for the results presented in this paper are archived at https://doi.org/10.5281/zenodo.1414423.

**Appendix A: Model selection and performance**

We fit the likelihood presented in Equation (1) with five different functional forms for $\mu_{CTW}$ (Equations (A1) - (A5)) and two different functional forms for $\sigma_{CTW}$ (Equations (A6) and (A7)), resulting in data from a total of 10 emulator models (each with different likelihoods based on $\mu_{CTW}, \sigma_{CTW}$) to compare to the C3MP data set.

The five functional forms for $\mu$ were selected intentionally. The first (Equation (A1)) is a second order Taylor polynomial

approximation of mean yield response. Equation (A2) is the functional form for mean response used in Ruane et al. (2014), differing from the second order Taylor polynomial by only one third-order CTW interaction term, $a_{10}$. Equations (A3) and (A4) continue to build up from the second order Taylor polynomial, examining the impacts of adding third order CTW interaction terms and the impacts of adding pure third order CTW terms respectively. Finally, Equation (A5) is the full third order Taylor

polynomial, a flexible approximation for many complicated functions. The two functional forms for $\sigma$ (Equations (A6) and (A7)) are simply the second and third order Taylor polynomial approximations of response spread across C3MP sites.

**quadratic:** $\mu_{CTW} = a_1 \Delta T + a_2 (\Delta T)^2 + a_3 \Delta W + a_4 (\Delta W)^2 + a_5 \Delta C + a_6 (\Delta C)^2 + a_7 \Delta T \Delta W + a_8 \Delta T \Delta C + a_9 \Delta W \Delta C$

$$\tag{A1}$$

**C3MP:** $\mu_{CTW} = a_1 \Delta T + a_2 (\Delta T)^2 + a_3 \Delta W + a_4 (\Delta W)^2 + a_5 \Delta C + a_6 (\Delta C)^2 + a_7 \Delta T \Delta W + a_8 \Delta T \Delta C + a_9 \Delta W \Delta C$

$\qquad + a_{10} \Delta T \Delta W \Delta C$

$$\tag{A2}$$

**cross:** $\mu_{CTW} = a_1 \Delta T + a_2 (\Delta T)^2 + a_3 \Delta W + a_4 (\Delta W)^2 + a_5 \Delta C + a_6 (\Delta C)^2 + a_7 \Delta T \Delta W + a_8 \Delta T \Delta C + a_9 \Delta W \Delta C$

$\qquad + a_{10} \Delta T \Delta W \Delta C$

$\qquad + a_{11} (\Delta T)^2 \Delta W + a_{12} (\Delta T)^2 \Delta C + a_{13} \Delta T (\Delta W)^2 + a_{14} \Delta T (\Delta C)^2 + a_{15} (\Delta W)^2 \Delta C + a_{16} \Delta W (\Delta C)^2$

$$\tag{A3}$$

**pure:** $\mu_{CTW} = a_1 \Delta T + a_2 (\Delta T)^2 + a_3 \Delta W + a_4 (\Delta W)^2 + a_5 \Delta C + a_6 (\Delta C)^2 + a_7 \Delta T \Delta W + a_8 \Delta T \Delta C + a_9 \Delta W \Delta C$

$\qquad + a_{10} (\Delta T)^3 + a_{11} (\Delta W)^3 + a_{12} (\Delta C)^3$

$$\tag{A4}$$

**cubic:** $\mu_{CTW} = a_1 \Delta T + a_2 (\Delta T)^2 + a_3 \Delta W + a_4 (\Delta W)^2 + a_5 \Delta C + a_6 (\Delta C)^2 + a_7 \Delta T \Delta W + a_8 \Delta T \Delta C + a_9 \Delta W \Delta C$

$\qquad + a_{10} \Delta T \Delta W \Delta C$

$\qquad + a_{11} (\Delta T)^2 \Delta W + a_{12} (\Delta T)^2 \Delta C + a_{13} \Delta T (\Delta W)^2 + a_{14} \Delta T (\Delta C)^2 + a_{15} (\Delta W)^2 \Delta C + a_{16} \Delta W (\Delta C)^2$

$\qquad + a_{17} (\Delta T)^3 + a_{18} (\Delta W)^3 + a_{19} (\Delta C)^3$

$$\tag{A5}$$

**quadratic:** $\sigma_{CTW} = |b_0 + b_1 \Delta T + b_2 (\Delta T)^2 + b_3 \Delta W + b_4 (\Delta W)^2 + b_5 \Delta C + b_6 (\Delta C)^2$

$\qquad + b_7 \Delta T \Delta W + b_8 \Delta T \Delta C + b_9 \Delta W \Delta C|$

$$\tag{A6}$$

**cubic:** $\sigma_{CTW} = |b_0 + b_1\Delta T + b_2(\Delta T)^2 + b_3\Delta W + b_4(\Delta W)^2 + b_5\Delta C + b_6(\Delta C)^2$

$$+ b_7\Delta T\Delta W + b_8\Delta T\Delta C + b_9\Delta W\Delta C + b_{10}\Delta T\Delta W\Delta C$$

$$+ b_{11}(\Delta T)^2\Delta W + b_{12}(\Delta T)^2\Delta C + b_{13}\Delta T(\Delta W)^2 + b_{14}\Delta T(\Delta C)^2 \tag{A7}$$

$$+ b_{15}(\Delta W)^2\Delta C + b_{16}\Delta W(\Delta C)^2 + b_{17}(\Delta T)^3 + b_{18}(\Delta W)^3 + b_{19}(\Delta C)^3|$$

We selected the model presented in the paper from the 10 combinations above based on leave-one (CTW test)-out cross-validation experiments to estimate out-of-sample prediction error for each production group. We do also include the in-sample performance metric defined in Section 3.1 for a more complete picture of model performance for all 10 functional form combinations for all 25 production groups (Tables A1-A9).

First, to test each model's validity and robustness at predicting yield changes for CTW values not included in the training data for each group, we ran leave-one-out cross-validation experiments (Gelman et al., 2014) to analyze the performance of each model for each production group. For each group separately, one CTW test data was withheld and the model was fit on the remaining 98 CTW tests. Then the mean, high, and low response functions resulting from the model were evaluated on the C3MP site data for the withheld test. This process was repeated withholding each CTW test, and the results were averaged resulting in an RMSE measure of performance for each of the mean, high, and low response functions. Leave-on-out cross validation used in this way answers the question: For a particular production group and model, on average, how do the emulated [mean, high, low] yield changes compare against the C3MP [mean, high, low] yield changes for CTW values not in the training set?

The Latin Hypercube design of the C3MP sensitivity tests lends confidence to this leave-one-out exercise because the cross-validation has covered the full space of CTW combinations. The results are summarized in Figure A1: each row represents the average leave-one-out cross-validation RMSE measures for each functional form across all production groups for the high, low, or mean response function, and then the average across all three (total, bottom row Figure A1). We find that cubic $\mu_{CTW}$, quadratic $\sigma_{CTW}$ performs the best at this cross validation experiment for the highest number of ensembles across the three response functions we defined in Equation (8) (that is, the high, low, and mean response functions). We repeat these calculations for each production group separately (rather than averaging across production groups) to determine the group-specific optimal functional form, listed in Table A10 for each group.

Because cubic $\mu_{CTW}$, quadratic $\sigma_{CTW}$ performs the best at out-of-sample error measurements for the highest number of ensembles across mean, low, and high response functions, and is quite good (though not the best) at in-sample error measurements (Table 1), this is the form used throughtout the body of the paper as the most broadly optimal functional form combination. We particularly value performance on the cross-validation (out-of-sample error) experiments because most CTW changes that may arise in application are likely to differ from the 99 C3MP tests.

We also repeat the in-sample measurement of error presented in Section 3.1 for all functional form combinations. These results are summarized in Tables A1 to A9, and we find that, purely based on the in-sample measurements, cubic $\mu_{CTW}$, cubic $\sigma_{CTW}$ (Table A9) is the best functional form for the most production groups. Specifically, it only performs poorly for

one crop, rainfed wheat in the mid-latitudes. However, it is very poor for that important ensemble. The in-sample performance information from these tables is included in Table A10 for each production-group specific optimal functional form combination.

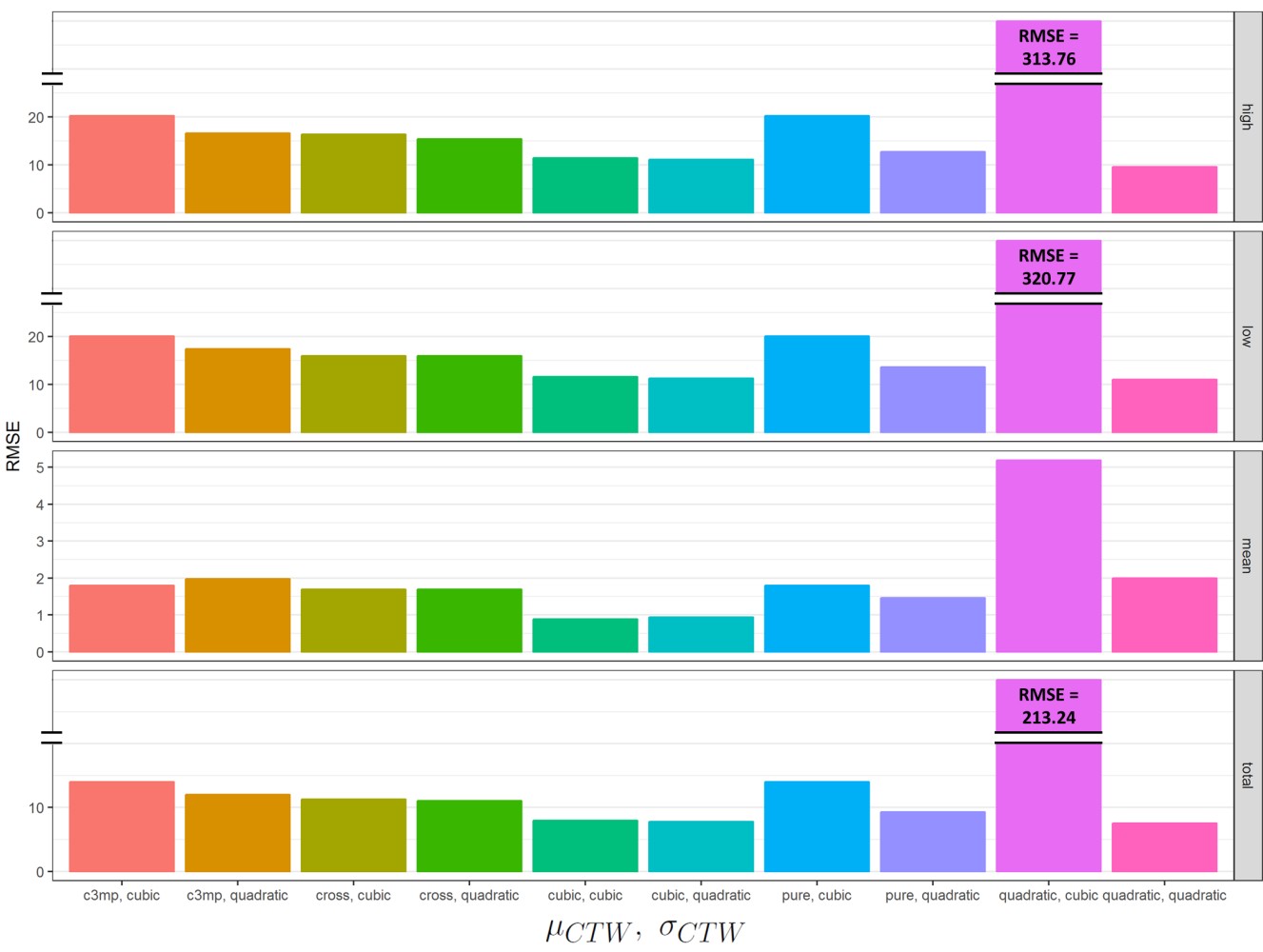

**Figure A1.** Comparison of leave-one-out cross-validation average RMSE measures for each functional form across all ensembles. Each functional form is labeled as $\mu_{CTW}$, $\sigma_{CTW}$. Note the broken scales to capture the performance of quadratic $\mu_{CTW}$, cubic $sigma_{CTW}$.

**Table A1.** Persephone v1.0 response function performance for all production groups, for quadratic $\mu_{CTW}$ (Equation (A1)), quadratic $\sigma_{CTW}$ (Equation (A6))

| Production group[1] | NRMS mean[2] | NRMS high | NRMS low | In-sample Performance |
|---|---|---|---|---|
| c3 IRR mid | 0.043 | 0.445 | 0.279 | Good |
| c3 IRR tropic | 0.074 | 0.270 | 0.188 | Good |
| c3 RFD mid | 0.044 | 0.301 | 0.280 | Good |
| c3 RFD tropic | 0.046 | 0.178 | 0.170 | Excellent |
| c4 IRR mid | 0.028 | 0.137 | 0.125 | Excellent |
| c4 IRR tropic | 0.041 | 1.662 | 0.440 | Poor |
| c4 RFD mid | 0.049 | 0.295 | 0.258 | Good |
| c4 RFD tropic | 0.094 | 0.280 | 0.209 | Good |
| Maize IRR mid | 0.028 | 0.150 | 0.130 | Excellent |
| Maize IRR tropic | 0.102 | 0.755 | 0.331 | Adequate |
| Maize RFD mid | 0.045 | 0.266 | 0.251 | Good |
| Maize RFD tropic | 0.108 | 0.318 | 0.188 | Good |
| Rice IRR mid | 0.069 | 0.259 | 0.181 | Good |
| Rice IRR tropic | 0.095 | 0.296 | 0.203 | Good |
| Rice RFD mid | 0.180 | 1.429 | 1.601 | Poor |
| Rice RFD tropic | 0.047 | 0.116 | 0.144 | Excellent |
| Soybeans IRR mid | 0.080 | 0.245 | 0.192 | Excellent |
| Soybeans IRR tropic | 0.068 | 0.119 | 0.175 | Excellent |
| Soybeans RFD mid | 0.069 | 0.139 | 0.178 | Excellent |
| Soybeans RFD tropic | 0.101 | 0.183 | 0.179 | Excellent |
| Sugarcane RFD tropic | 0.125 | 0.448 | 0.479 | Good |
| Wheat IRR mid | 0.069 | 0.408 | 0.351 | Good |
| Wheat IRR tropic | 0.085 | 0.309 | 0.267 | Good |
| Wheat RFD mid | 0.041 | 0.298 | 0.286 | Good |
| Wheat RFD tropic | 0.199 | 0.807 | 0.833 | Adequate |

1. "IRR" = irrigated, "RFD" = rainfed, "mid" = mid-latitudes (30- 70°S, 30- 70°N), "tropic" = 30°S to 30°N. 2. Note that the mean response function performs "excellent" for all production groups.

**Table A2.** Persephone v1.0 response function performance for all production groups, for quadratic $\mu_{CTW}$ (Equation (A1)), cubic $\sigma_{CTW}$ (Equation (A7))

| Production group[1] | NRMS mean[2] | NRMS high | NRMS low | In-sample Performance |
|---|---|---|---|---|
| c3 IRR mid | 0.036 | 0.142 | 0.110 | Excellent |
| c3 IRR tropic | 0.074 | 0.661 | 0.525 | Adequate |
| c3 RFD mid | 0.044 | 0.899 | 0.706 | Adequate |
| c3 RFD tropic | 0.052 | 0.817 | 0.571 | Adequate |
| c4 IRR mid | 0.028 | 2.169 | 0.863 | Poor |
| c4 IRR tropic | 0.035 | 0.300 | 0.063 | Good |
| c4 RFD mid | 0.042 | 0.125 | 0.080 | Excellent |
| c4 RFD tropic | 0.084 | 1.022 | 0.511 | Poor |
| Maize IRR mid | 0.035 | 0.763 | 0.471 | Adequate |
| Maize IRR tropic | 0.083 | 0.193 | 0.066 | Excellent |
| Maize RFD mid | 0.039 | 0.112 | 0.075 | Excellent |
| Maize RFD tropic | 0.086 | 0.390 | 0.147 | Good |
| Rice IRR mid | 0.064 | 0.159 | 0.098 | Excellent |
| Rice IRR tropic | 0.095 | 1.029 | 0.672 | Poor |
| Rice RFD mid | 0.153 | 0.166 | 0.187 | Excellent |
| Rice RFD tropic | 0.047 | 0.077 | 0.063 | Excellent |
| Soybeans IRR mid | 0.073 | 0.123 | 0.088 | Excellent |
| Soybeans IRR tropic | 0.057 | 0.078 | 0.089 | Excellent |
| Soybeans RFD mid | 0.075 | 1.137 | 0.893 | Poor |
| Soybeans RFD tropic | 0.084 | 0.355 | 0.303 | Excellent |
| Sugarcane RFD tropic | 0.114 | 2.163 | 1.469 | Poor |
| Wheat IRR mid | 0.060 | 0.149 | 0.197 | Excellent |
| Wheat IRR tropic | 0.082 | 0.206 | 0.204 | Excellent |
| Wheat RFD mid | 0.038 | 0.117 | 0.102 | Excellent |
| Wheat RFD tropic | 0.175 | 0.769 | 0.587 | Adequate |

1. "IRR" = irrigated, "RFD" = rainfed, "mid" = mid-latitudes (30- 70°S, 30- 70°N), "tropic" = 30°S to 30°N. 2. Note that the mean response function performs "excellent" for all production groups.

**Table A3.** Persephone v1.0 response function performance for all production groups, for C3MP $\mu_{CTW}$ (Equation (A2)), quadratic $\sigma_{CTW}$ (Equation (A6))

| Production group[1] | NRMS mean[2] | NRMS high | NRMS low | In-sample Performance |
|---|---|---|---|---|
| c3 IRR mid | 0.037 | 1.039 | 0.7 | Poor |
| c3 IRR tropic | 0.074 | 1.675 | 0.792 | Poor |
| c3 RFD mid | 0.046 | 0.303 | 0.276 | Good |
| c3 RFD tropic | 0.057 | 1.116 | 0.78 | Poor |
| c4 IRR mid | 0.027 | 0.139 | 0.123 | Excellent |
| c4 IRR tropic | 0.049 | 0.894 | 0.224 | Adequate |
| c4 RFD mid | 0.046 | 0.303 | 0.248 | Good |
| c4 RFD tropic | 0.093 | 0.3 | 0.199 | Good |
| Maize IRR mid | 0.027 | 0.152 | 0.129 | Excellent |
| Maize IRR tropic | 0.111 | 1.091 | 0.273 | Poor |
| Maize RFD mid | 0.042 | 0.272 | 0.242 | Good |
| Maize RFD tropic | 0.106 | 0.341 | 0.182 | Good |
| Rice IRR mid | 0.081 | 0.725 | 0.402 | Adequate |
| Rice IRR tropic | 0.093 | 0.287 | 0.209 | Good |
| Rice RFD mid | 0.115 | 1.055 | 1.08 | Poor |
| Rice RFD tropic | 0.047 | 0.18 | 0.164 | Excellent |
| Soybeans IRR mid | 0.08 | 0.248 | 0.191 | Excellent |
| Soybeans IRR tropic | 0.11 | 0.726 | 0.724 Adequate | |
| Soybeans RFD mid | 0.066 | 0.149 | 0.157 | Excellent |
| Soybeans RFD tropic | 0.084 | 0.444 | 0.354 | Good |
| Sugarcane RFD tropic | 0.144 | 2.42 | 2.066 | Poor |
| Wheat IRR mid | 0.061 | 0.391 | 0.365 | Good |
| Wheat IRR tropic | 0.082 | 0.72 | 0.548 | Adequate |
| Wheat RFD mid | 0.041 | 0.298 | 0.287 | Good |
| Wheat RFD tropic | 0.147 | 0.297 | 0.376 | Good |

1. "IRR" = irrigated, "RFD" = rainfed, "mid" = mid-latitudes (30- 70°S, 30- 70°N), "tropic" = 30°S to 30°N. 2. Note that the mean response function performs "excellent" for all production groups.

**Table A4.** Persephone v1.0 response function performance for all production groups, for C3MP $\mu_{CTW}$ (Equation (A2)), cubic $\sigma_{CTW}$ (Equation (A7))

| Production group[1] | NRMS mean[2] | NRMS high | NRMS low | In-sample Performance |
|---|---|---|---|---|
| c3 IRR mid | 0.032 | 0.356 | 0.240 | Good |
| c3 IRR tropic | 0.073 | 0.113 | 0.113 | Excellent |
| c3 RFD mid | 0.039 | 0.121 | 0.094 | Excellent |
| c3 RFD tropic | 0.041 | 0.087 | 0.057 | Excellent |
| c4 IRR mid | 0.026 | 0.166 | 0.108 | Excellent |
| c4 IRR tropic | 0.037 | 0.296 | 0.064 | Good |
| c4 RFD mid | 0.038 | 0.449 | 0.358 | Good |
| c4 RFD tropic | 0.073 | 0.335 | 0.168 | Good |
| Maize IRR mid | 0.025 | 0.073 | 0.044 | Excellent |
| Maize IRR tropic | 0.082 | 0.244 | 0.082 | Good |
| Maize RFD mid | 0.036 | 0.109 | 0.076 | Excellent |
| Maize RFD tropic | 0.096 | 0.729 | 0.272 | Adequate |
| Rice IRR mid | 0.064 | 0.282 | 0.175 | Good |
| Rice IRR tropic | 0.094 | 0.120 | 0.143 | Excellent |
| Rice RFD mid | 0.134 | 0.175 | 0.178 | Excellent |
| Rice RFD tropic | 0.046 | 0.079 | 0.060 | Excellent |
| Soybeans IRR mid | 0.073 | 0.123 | 0.088 | Excellent |
| Soybeans IRR tropic | 0.075 | 0.213 | 0.194 | Excellent |
| Soybeans RFD mid | 0.060 | 0.080 | 0.068 | Excellent |
| Soybeans RFD tropic | 0.086 | 0.145 | 0.169 | Excellent |
| Sugarcane RFD tropic | 0.111 | 0.175 | 0.100 | Excellent |
| Wheat IRR mid | 0.061 | 0.961 | 1.039 | Poor |
| Wheat IRR tropic | 0.088 | 2.522 | 1.231 | Poor |
| Wheat RFD mid | 0.058 | 7.604 | 2.233 | Poor |
| Wheat RFD tropic | 0.164 | 0.934 | 0.924 | Adequate |

1. "IRR" = irrigated, "RFD" = rainfed, "mid" = mid-latitudes (30- 70°S, 30- 70°N), "tropic" = 30°S to 30°N. 2. Note that the mean response function performs "excellent" for all production groups.

**Table A5.** Persephone v1.0 response function performance for all production groups, for cross $\mu_{CTW}$ (Equation (A3)), quadratic $\sigma_{CTW}$ (Equation (A6))

| Production group[1] | NRMS mean[2] | NRMS high | NRMS low | In-sample Performance |
|---|---|---|---|---|
| c3 IRR mid | 0.022 | 1.038 | 0.701 | Poor |
| c3 IRR tropic | 0.073 | 1.671 | 0.792 | Poor |
| c3 RFD mid | 0.021 | 0.314 | 0.272 | Good |
| c3 RFD tropic | 0.030 | 1.201 | 0.634 | Poor |
| c4 IRR mid | 0.024 | 0.140 | 0.121 | Excellent |
| c4 IRR tropic | 0.041 | 0.928 | 0.220 | Poor |
| c4 RFD mid | 0.033 | 0.312 | 0.247 | Good |
| c4 RFD tropic | 0.069 | 0.340 | 0.187 | Good |
| Maize IRR mid | 0.025 | 0.152 | 0.128 | Excellent |
| Maize IRR tropic | 0.107 | 1.926 | 0.450 | Poor |
| Maize RFD mid | 0.030 | 0.286 | 0.236 | Good |
| Maize RFD tropic | 0.083 | 0.379 | 0.175 | Good |
| Rice IRR mid | 0.070 | 0.627 | 0.445 | Poor |
| Rice IRR tropic | 0.092 | 0.347 | 0.258 | Good |
| Rice RFD mid | 0.092 | 0.306 | 0.342 | Good |
| Rice RFD tropic | 0.020 | 0.210 | 0.141 | Excellent |
| Soybeans IRR mid | 0.090 | 1.595 | 1.103 | Poor |
| Soybeans IRR tropic | 0.051 | 0.203 | 0.161 | Excellent |
| Soybeans RFD mid | 0.036 | 0.150 | 0.148 | Excellent |
| Soybeans RFD tropic | 0.081 | 0.318 | 0.219 | Good |
| Sugarcane RFD tropic | 0.147 | 5.574 | 3.954 | Poor |
| Wheat IRR mid | 0.056 | 0.392 | 0.364 | Good |
| Wheat IRR tropic | 0.078 | 1.256 | 0.815 | Poor |
| Wheat RFD mid | 0.034 | 0.306 | 0.279 | Good |
| Wheat RFD tropic | 0.114 | 0.332 | 0.347 | Good |

1. "IRR" = irrigated, "RFD" = rainfed, "mid" = mid-latitudes (30- 70°S, 30- 70°N), "tropic" = 30°S to 30°N. 2. Note that the mean response function performs "excellent" for all production groups.

**Table A6.** Persephone v1.0 response function performance for all production groups, for cross $\mu_{CTW}$ (Equation (A3)), cubic $\sigma_{CTW}$ (Equation (A7))

| Production group[1] | NRMS mean[2] | NRMS high | NRMS low | In-sample Performance |
|---|---|---|---|---|
| c3 IRR mid | 0.019 | 0.303 | 0.196 | Good |
| c3 IRR tropic | 0.071 | 0.112 | 0.111 | Excellent |
| c3 RFD mid | 0.022 | 0.674 | 0.602 | Adequate |
| c3 RFD tropic | 0.025 | 0.071 | 0.056 | Excellent |
| c4 IRR mid | 0.024 | 0.168 | 0.114 | Excellent |
| c4 IRR tropic | 0.032 | 0.303 | 0.076 | Good |
| c4 RFD mid | 0.037 | 1.544 | 0.623 | Poor |
| c4 RFD tropic | 0.062 | 0.156 | 0.060 | Excellent |
| Maize IRR mid | 0.022 | 0.071 | 0.044 | Excellent |
| Maize IRR tropic | 0.074 | 0.179 | 0.063 | Excellent |
| Maize RFD mid | 0.028 | 0.097 | 0.081 | Excellent |
| Maize RFD tropic | 0.073 | 0.305 | 0.129 | Good |
| Rice IRR mid | 0.063 | 0.278 | 0.176 | Good |
| Rice IRR tropic | 0.092 | 0.120 | 0.141 | Excellent |
| Rice RFD mid | 0.096 | 0.237 | 0.219 | Good |
| Rice RFD tropic | 0.019 | 0.057 | 0.051 | Excellent |
| Soybeans IRR mid | 0.058 | 0.120 | 0.073 | Excellent |
| Soybeans IRR tropic | 0.063 | 0.120 | 0.212 | Excellent |
| Soybeans RFD mid | 0.034 | 0.054 | 0.054 | Excellent |
| Soybeans RFD tropic | 0.053 | 0.111 | 0.094 | Excellent |
| Sugarcane RFD tropic | 0.078 | 0.241 | 0.229 | Excellent |
| Wheat IRR mid | 0.044 | 0.721 | 0.748 | Adequate |
| Wheat IRR tropic | 0.084 | 0.185 | 0.219 | Excellent |
| Wheat RFD mid | 0.050 | 3.658 | 2.116 | Poor |
| Wheat RFD tropic | 0.111 | 0.212 | 0.179 | Excellent |

1. "IRR" = irrigated, "RFD" = rainfed, "mid" = mid-latitudes (30- 70°S, 30- 70°N), "tropic" = 30°S to 30°N. 2. Note that the mean response function performs "excellent" for all production groups.

**Table A7.** Persephone v1.0 response function performance for all production groups, for pure $\mu_{CTW}$ (Equation (A4)), quadratic $\sigma_{CTW}$ (Equation (A6))

| Production group[1] | NRMS mean[2] | NRMS high | NRMS low | In-sample Performance |
|---|---|---|---|---|
| c3 IRR mid | 0.031 | 1.045 | 0.697 | Poor |
| c3 IRR tropic | 0.071 | 1.660 | 0.791 | Poor |
| c3 RFD mid | 0.039 | 0.301 | 0.280 | Good |
| c3 RFD tropic | 0.052 | 0.662 | 0.498 | Poor |
| c4 IRR mid | 0.012 | 0.149 | 0.111 | Excellent |
| c4 IRR tropic | 0.018 | 0.985 | 0.216 | Poor |
| c4 RFD mid | 0.035 | 0.307 | 0.248 | Good |
| c4 RFD tropic | 0.045 | 0.334 | 0.189 | Good |
| Maize IRR mid | 0.012 | 0.165 | 0.117 | Excellent |
| Maize IRR tropic | 0.016 | 1.039 | 0.340 | Poor |
| Maize RFD mid | 0.035 | 0.277 | 0.242 | Good |
| Maize RFD tropic | 0.044 | 0.376 | 0.179 | Good |
| Rice IRR mid | 0.038 | 0.346 | 0.197 | Good |
| Rice IRR tropic | 0.091 | 0.343 | 0.260 | Good |
| Rice RFD mid | 0.124 | 0.123 | 0.275 | Good |
| Rice RFD tropic | 0.053 | 0.161 | 0.171 | Excellent |
| Soybeans IRR mid | 0.033 | 0.221 | 0.185 | Excellent |
| Soybeans IRR tropic | 0.066 | 0.072 | 0.172 | Excellent |
| Soybeans RFD mid | 0.056 | 0.137 | 0.170 | Excellent |
| Soybeans RFD tropic | 0.083 | 0.171 | 0.173 | Excellent |
| Sugarcane RFD tropic | 0.085 | 1.504 | 1.307 | Poor |
| Wheat IRR mid | 0.045 | 0.378 | 0.377 | Good |
| Wheat IRR tropic | 0.080 | 0.710 | 0.550 | Good |
| Wheat RFD mid | 0.034 | 0.294 | 0.289 | Good |
| Wheat RFD tropic | 0.175 | 0.371 | 0.341 | Good |

1. "IRR" = irrigated, "RFD" = rainfed, "mid" = mid-latitudes (30- 70°S, 30- 70°N), "tropic" = 30°S to 30°N. 2. Note that the mean response function performs "excellent" for all production groups.

**Table A8.** Persephone v1.0 response function performance for all production groups, for pure $\mu_{CTW}$ (Equation (A4)), cubic $\sigma_{CTW}$ (Equation (A7))

| Production group[1] | NRMS mean[2] | NRMS high | NRMS low | In-sample Performance |
|---|---|---|---|---|
| c3 IRR mid | 0.030 | 0.766 | 0.524 | Adequate |
| c3 IRR tropic | 0.071 | 0.115 | 0.110 | Excellent |
| c3 RFD mid | 0.035 | 0.117 | 0.095 | Excellent |
| c3 RFD tropic | 0.040 | 0.082 | 0.061 | Excellent |
| c4 IRR mid | 0.012 | 0.153 | 0.116 | Excellent |
| c4 IRR tropic | 0.013 | 0.249 | 0.072 | Excellent |
| c4 RFD mid | 0.038 | 2.286 | 0.778 | Poor |
| c4 RFD tropic | 0.040 | 0.120 | 0.061 | Excellent |
| Maize IRR mid | 0.012 | 0.061 | 0.046 | Excellent |
| Maize IRR tropic | 0.016 | 0.162 | 0.073 | Excellent |
| Maize RFD mid | 0.031 | 0.104 | 0.077 | Excellent |
| Maize RFD tropic | 0.041 | 0.126 | 0.060 | Excellent |
| Rice IRR mid | 0.038 | 0.109 | 0.076 | Excellent |
| Rice IRR tropic | 0.092 | 0.123 | 0.139 | Excellent |
| Rice RFD mid | 0.122 | 0.178 | 0.213 | Excellent |
| Rice RFD tropic | 0.043 | 0.213 | 0.149 | Excellent |
| Soybeans IRR mid | 0.029 | 0.091 | 0.071 | Excellent |
| Soybeans IRR tropic | 0.065 | 0.125 | 0.141 | Excellent |
| Soybeans RFD mid | 0.052 | 0.072 | 0.061 | Excellent |
| Soybeans RFD tropic | 0.066 | 0.112 | 0.105 | Excellent |
| Sugarcane RFD tropic | 0.066 | 0.260 | 0.177 | Good |
| Wheat IRR mid | 0.033 | 0.691 | 0.705 | Adequate |
| Wheat IRR tropic | 0.078 | 0.185 | 0.215 | Excellent |
| Wheat RFD mid | 0.037 | 5.732 | 2.313 | Poor |
| Wheat RFD tropic | 0.173 | 0.368 | 0.204 | Good |

1. "IRR" = irrigated, "RFD" = rainfed, "mid" = mid-latitudes (30- 70°S, 30- 70°N), "tropic" = 30°S to 30°N. 2. Note that the mean response function performs "excellent" for all production groups.

**Table A9.** Persephone v1.0 response function performance for all production groups, for cubic $\mu_{CTW}$ (Equation (A5)), cubic $\sigma_{CTW}$ (Equation (A7))

| Production group[1] | NRMS mean[2] | NRMS high | NRMS low | In-sample Performance |
|---|---|---|---|---|
| c3 IRR mid | 0.013 | 0.488 | 0.326 | Good |
| c3 IRR tropic | 0.069 | 0.113 | 0.109 | Excellent |
| c3 RFD mid | 0.009 | 0.106 | 0.095 | Excellent |
| c3 RFD tropic | 0.021 | 0.065 | 0.058 | Excellent |
| c4 IRR mid | 0.010 | 0.152 | 0.116 | Excellent |
| c4 IRR tropic | 0.010 | 0.313 | 0.092 | Good |
| c4 RFD mid | 0.016 | 0.705 | 0.370 | Adequate |
| c4 RFD tropic | 0.018 | 0.102 | 0.058 | Excellent |
| Maize IRR mid | 0.010 | 0.062 | 0.044 | Excellent |
| Maize IRR tropic | 0.011 | 0.116 | 0.066 | Excellent |
| Maize RFD mid | 0.016 | 0.091 | 0.079 | Excellent |
| Maize RFD tropic | 0.021 | 0.109 | 0.056 | Excellent |
| Rice IRR mid | 0.029 | 0.104 | 0.073 | Excellent |
| Rice IRR tropic | 0.089 | 0.123 | 0.137 | Excellent |
| Rice RFD mid | 0.043 | 0.098 | 0.123 | Excellent |
| Rice RFD tropic | 0.018 | 0.060 | 0.048 | Excellent |
| Soybeans IRR mid | 0.015 | 0.087 | 0.068 | Excellent |
| Soybeans IRR tropic | 0.034 | 0.063 | 0.085 | Excellent |
| Soybeans RFD mid | 0.015 | 0.042 | 0.046 | Excellent |
| Soybeans RFD tropic | 0.035 | 0.100 | 0.089 | Excellent |
| Sugarcane RFD tropic | 0.042 | 0.209 | 0.171 | Excellent |
| Wheat IRR mid | 0.022 | 0.681 | 0.675 | Adequate |
| Wheat IRR tropic | 0.078 | 0.171 | 0.221 | Excellent |
| Wheat RFD mid | 0.042 | 5.268 | 1.905 | Poor |
| Wheat RFD tropic | 0.091 | 0.196 | 0.165 | Excellent |

1. "IRR" = irrigated, "RFD" = rainfed, "mid" = mid-latitudes (30- $70^\circ$ S, 30- $70^\circ$ N), "tropic" = $30^\circ$ S to $30^\circ$ N. 2. Note that the mean response function performs "excellent" for all production groups.

**Table A10.** The best performing functional form combination for each production group at the task of leave-one-out cross-validation(out of sample performance) and the corresponding In-sample Performance measure.

| Production group | $\mu_{CTW}$ | $\sigma_{CTW}$ | In-sample Performance |
|---|---|---|---|
| c3 IRR mid | quadratic | quadratic | Good |
| c3 IRR tropic | cubic | cubic | Excellent |
| c3 RFD mid | c3mp | cubic | Excellent |
| c3 RFD tropic | cubic | cubic | Excellent |
| c4 IRR mid | cubic | cubic | Excellent |
| c4 IRR tropic | cubic | cubic | Good |
| c4 RFD mid | cubic | quadratic | Good |
| c4 RFD tropic | pure | quadratic | Good |
| Maize IRR mid | cubic | cubic | Excellent |
| Maize IRR tropic | cubic | cubic | Excellent |
| Maize RFD mid | cubic | cubic | Excellent |
| Maize RFD tropic | cubic | quadratic | Good |
| Rice IRR mid | cubic | cubic | Excellent |
| Rice IRR tropic | quadratic | quadratic | Good |
| Rice RFD mid | cubic | cubic | Excellent |
| Rice RFD tropic | cubic | cubic | Excellent |
| Soybeans IRR mid | cubic | cubic | Excellent |
| Soybeans IRR tropic | quadratic | cubic | Excellent |
| Soybeans RFD mid | cubic | cubic | Excellent |
| Soybeans RFD tropic | cubic | cubic | Excellent |
| Sugarcane RFD tropic | c3mp | cubic | Excellent |
| Wheat IRR mid | quadratic | cubic | Excellent |
| Wheat IRR tropic | quadratic | quadratic | Good |
| Wheat RFD mid | cubic | quadratic | Good |
| Wheat RFD tropic | cubic | cubic | Excellent |

## Appendix B: C3MP baseline yield estimate functional forms

As mentioned in Section 2.1.1, the 8 different functional forms used to relate site-specific output yield in response to input CTW values are presented here in Equations (B1)-(B8). Each functional form was used with each specific C3MP site's data in order to provide a best estimate of baseline yield for that site. The form with the smallest root mean square error across the 99 tests for the site is the one used to provide a best estimate of baseline yield. This best estimate of baseline yield is used to convert the C3MP output yields at the site to percent changes in yield from baseline for emulator training.

$$Y_{CTW}^{site} = a_0 + a_1 \Delta T + a_2 \Delta W + a_3 \Delta C \tag{B1}$$

$$Y_{CTW}^{site} = a_0 + a_1 \Delta T + a_2 (\Delta T)^2 + a_3 \Delta W + a_4 (\Delta W)^2 + a_5 \Delta C + a_6 (\Delta C)^2 \tag{B2}$$

$$Y_{CTW}^{site} = a_0 + a_1 \Delta T + a_2 \Delta W + a_3 \Delta C + a_4 \Delta T \Delta W + a_5 \Delta T \Delta C + a_6 \Delta W \Delta C \tag{B3}$$

$$Y_{CTW}^{site} = a_0 + a_1 \Delta T + a_2 (\Delta T)^2 + a_3 \Delta W + a_4 (\Delta W)^2 + a_5 \Delta C + a_6 (\Delta C)^2 + a_7 \Delta T \Delta W + a_8 \Delta T \Delta C + a_9 \Delta W \Delta C \tag{B4}$$

$$\begin{aligned} Y_{CTW}^{site} = {} & a_0 + a_1 \Delta T + a_2 (\Delta T)^2 + a_3 \Delta W + a_4 (\Delta W)^2 + a_5 \Delta C + a_6 (\Delta C)^2 + a_7 \Delta T \Delta W + a_8 \Delta T \Delta C + a_9 \Delta W \Delta C \\ & + a_{10} \Delta T \Delta W \Delta C \end{aligned} \tag{B5}$$

$$\begin{aligned} Y_{CTW}^{site} = {} & a_0 + a_1 \Delta T + a_2 (\Delta T)^2 + a_3 \Delta W + a_4 (\Delta W)^2 + a_5 \Delta C + a_6 (\Delta C)^2 + a_7 \Delta T \Delta W + a_8 \Delta T \Delta C + a_9 \Delta W \Delta C \\ & + a_{10} \Delta T \Delta W \Delta C \\ & + a_{11} (\Delta T)^2 \Delta W + a_{12} (\Delta T)^2 \Delta C + a_{13} \Delta T (\Delta W)^2 + a_{14} \Delta T (\Delta C)^2 + a_{15} (\Delta W)^2 \Delta C + a_{16} \Delta W (\Delta C)^2 \end{aligned} \tag{B6}$$

$$\begin{aligned} Y_{CTW}^{site} = {} & a_0 + a_1 \Delta T + a_2 (\Delta T)^2 + a_3 \Delta W + a_4 (\Delta W)^2 + a_5 \Delta C + a_6 (\Delta C)^2 + a_7 \Delta T \Delta W + a_8 \Delta T \Delta C + a_9 \Delta W \Delta C \\ & + a_{10} (\Delta T)^3 + a_{11} (\Delta W)^3 + a_{12} (\Delta C)^3 \end{aligned} \tag{B7}$$

$$\begin{aligned} Y_{CTW}^{site} = {} & a_0 + a_1 \Delta T + a_2 (\Delta T)^2 + a_3 \Delta W + a_4 (\Delta W)^2 + a_5 \Delta C + a_6 (\Delta C)^2 + a_7 \Delta T \Delta W + a_8 \Delta T \Delta C + a_9 \Delta W \Delta C \\ & + a_{10} \Delta T \Delta W \Delta C \\ & + a_{11} (\Delta T)^2 \Delta W + a_{12} (\Delta T)^2 \Delta C + a_{13} \Delta T (\Delta W)^2 + a_{14} \Delta T (\Delta C)^2 + a_{15} (\Delta W)^2 \Delta C + a_{16} \Delta W (\Delta C)^2 \\ & + a_{17} (\Delta T)^3 + a_{18} (\Delta W)^3 + a_{19} (\Delta C)^3 \end{aligned} \tag{B8}$$

*Author contributions.* ACR and KC conceived the uses of this emulator. AS, MP, ACR, KC developed the emulator. AS wrote the manuscript with contributions from MP, KC, ACR.

*Competing interests.* No competing interests are present.

*Acknowledgements.* The authors appreciate discussions with Sonali McDermid and Theo Mavromatis around the evaluation and application
5   of the C3MP ensemble. Abigail Snyder, Katherine Calvin, and Meridel Phillips were supported by the U.S. Department of Energy, Office of Science, as part of research in the Multi-Sector Dynamics, Earth and Environmental System Modeling Program Area. Alex Ruane's work was supported by the NASA Climate Impacts Group under the Modeling, Analysis, and Prediction Program.

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
