# Peer review of "A crop yield change emulator for use in GCAM and similar models: Persephone v1.0"

_Geoscientific Model Development, 2018_

## Referee Comment (RC1) · Anonymous Referee #1 · 12 Nov 2018

General comments: The paper by Snyder et al. presents a crop yield change emulator for use in multi-sector economic models. The emulator is intended to support studies of the feedback loop among socioeconomics, Earth system changes, and crop yield changes, by providing multi-sector economic models with rapidly generated yield responses to Earth system changes, including some quantification of crop response uncertainty. Compared to using computationally expensive global gridded crop models to estimate yield changes, the emulator has the advantage of being more computationally efficient, allowing to run large ensembles of simulations necessary to explore future response options.

The work presented in the paper is highly relevant because it promises to allow new agricultural impact studies that do not only consider the impact of Earth system

changes on the agricultural system, but also how responses by society to cope with these changes may feed back to the Earth system. This bidirectional feedback has often been missing in previous impact assessments, or could only be analyzed for a small number of simulations due to its high computational requirements. While the emulator is developed for a specific multi-sector economic model, GCAM, it can also be used by other similar-in-scale models. The software implementing the emulator is published alongside the paper.

The paper is well written. The authors clearly describe the data processing and derivation of the emulator response functions (section 2) and their evaluation (section 3), finishing with a short example of an application of the emulator and some final conclusions and discussion. Although I have to admit that I am not very familiar with Bayesian statistics and found some of the corresponding terminology confusing I was still able to follow what was done.

I have two main concerns with the paper: the first refers to the C3MP dataset used to derive the emulator response functions, and the second refers to how these response functions are going to be used. The C3MP dataset is a large set of 99 CTW sensitivity tests carried out by a number of site-based crop models covering a range of different crops at a total of 1135 sites, of which data from 575 sites are actually used in this study. The climate change signal used in these sensitivity tests is completely synthetic since it consists of applying a temporally uniform temperature offset or precipitation multiplier to a historical baseline weather timeseries. However, in reality, climate change is not constant over time. For example, precipitation might increase during part of the year, while decreasing during other times. I am not convinced that the constant CTW perturbation experiments are equivalent to using a more realistic climate timeseries, and I would suggest that the authors either show in the paper or at least point the reader to other literature showing that this is a valid approach. Otherwise, the authors risk that there is already a bias in the data used to train the emulator which would propagate to the emulated yields.

[Figure]

My second concern is that response functions for some of the 25 production groups are based on a very small number of sites, in the most extreme case only two sites. I cannot help but wonder how representative these results really are, keeping in mind that each production group represents one crop-irrigation-latitude combination, with latitudes only distinguished into extended tropics and mid-latitudes. In addition, these sites are not only used to derive a mean yield response, but also an estimate of response uncertainty. The high and low response functions are supposed to represent site responses at the mean plus/minus one standard deviation level but how meaningful are these estimates based on such a small sample size? The emulator response functions are only derived for two regions (extended tropics and mid-latitudes) but I wonder at what level of spatial detail they will be used later. In Figure 8 and 9 of the paper, the authors take spatial patterns of climate change from the HadGEM2-ES model and use their response functions to derive corresponding patterns of yield change. These patterns are shown, but not evaluated in any way. I would suggest that the authors compare their derived patterns of yield change to simulations of yield change from global gridded crop models for the same climate data. After all, the intent of the emulator is to replace simulations by global gridded crop models. Such crop yield simulations for the HadGEM2-ES RCP8.5 scenario used in Figure 8 and 9 of the paper are, for example, available from a number of crop models and for a number of different crops within the ISIMIP data archive at https://esg.pik-potsdam.de/projects/isimip/ Such a comparison would help to address both of my concerns voiced above.

Specific comments:

- Page 2, ll. 25 – 28: You state that previous emulators were restricted to emulating yield change under RCP scenarios. I am aware of at least two other global crop yield change emulators derived from crop model simulations that are applicable to any future climate scenario: Oyebamiji et al. (2015) and Ostberg et al. (2018). It might be useful to contrast them to the work presented in this paper.

- Page 4, starting with line 11, and Figure 1: You outline three use cases for the Persephone yield emulator, but none of this is actually done in this paper. So I'm not sure if the Methods section is the right place for this.

- Page 6, section 2.2: The Copernicus guidelines request that datasets should be cited with a reference in the reference list. Is there a reference for the C3MP dataset?

- Page 6, ll. 8 – 14: This part does not refer to the setup of C3MP or to your processing of the C3MP data. Instead, it refers to how climate data is pre-processed before use with the finished emulator. It should probably be moved to section 4.

- Page 7, ll. 7 – 9: Unless I am mistaken, the global gridded crop models within AgMIP also conducted the 99 CTW sensitivity tests. They should offer a much better global coverage. Would it be worthwhile adding some discussion of why this paper used the site-based results instead of the global simulations?

- Page 8, ll. 19 – 20: Are the 8 different functional forms documented anywhere? Are these the same as are used in the emulator?

- Page 11, ll. 9 – 10: Is the value of b0 constrained between -0.02 and 0.02 or is it 0.02%? The following sentence suggests that it is 0.02%.

- Page 13, ll. 1 – 8: Given the very small sample sizes of about 1/3 of the production groups the 84.135th and 15.865th percentiles do not seem very meaningful

- Section 3.2: It seems to me that this is rather a test whether the crop models used in training feature these important relationships. Or would you say it is possible that these relationships are present in the models, but missing in the emulator? Since the evaluation is positive I guess it means that the relationships are present in the models and retained in the emulator.

- Page 20, l. 4 – page 21, l. 1: Climate change can affect both the start and the length of the growing season. Is this accounted for or is the same growing season used under climate change as during the reference period?

- Page 21, ll. 8 – 10: On the one hand you talk about passing CTW changes for regions into the emulator, on the other hand you mention a gridded map of yield changes. So are the yield changes at the same spatial resolution as the climate data or is there a difference between resolutions (region versus grid)? Please clarify.

- Page 21, ll. 17 – 18, and Figure 9: Looking at Figure 9, it seems to me that most regions show a positive yield change under the high response function, not just "a few regions". Also, I think that in the bottom row of Figure 9 the maps for mean and high response are swapped. The bottom right map (which should be the high response) looks identical to the map for Maize in Figure 8 (which shows the mean response).

- Page 23, ll. 6 – 8: Here, you emphasize the rapid evaluation time of the response functions relative to a global gridded crop model, but I think you should really try to show that the emulator response is actually comparable to what you would get using a global gridded crop model. This step is missing in the paper.

- Page 24, ll. 8 – 10: Given that the response functions are only derived for two latitudinal bands I would say that they cannot really be used to characterize the range of uncertainty within national or multi-national units (unless the respective unit is covered by C3MP sites).

- Page 24: In the paragraph on caveats, I would suggest to add discussion of potential biases arising out of way the CTW experiments are set up. Another source of uncertainty results from the fact that crop model simulations generally omit adaptation options such as changing sowing dates or switching to different cultivars. This is done for simplicity and comparability but is not realistic considering that agriculture is a highly managed system, and potentially creates another bias in the crop model simulations of yield change that are used to train the emulator.

Technical corrections:

- There are a number of typing errors throughout the paper (mostly missing characters,

sometimes missing words). Please check during the revision.

- The greyscale lines in Figure 6, 8 and 9 are hard to see at all, let alone distinguish the different shades of grey.

References:

Ostberg, Sebastian, Jacob Schewe, Katelin Childers, and Katja Frieler. 2018. "Changes in Crop Yields and Their Variability at Different Levels of Global Warming." Earth System Dynamics 9 (2): 479–96. doi:10.5194/esd-9-479-2018.

Oyebamiji, Oluwole K, Neil R Edwards, Philip B Holden, Paul H Garthwaite, Sibyll Schaphoff, and Dieter Gerten. 2015. "Emulating Global Climate Change Impacts on Crop Yields." Statistical Modelling 15 (6): 499–525. doi:10.1177/1471082X14568248.

---

## Referee Comment (RC2) · Anonymous Referee #2 · 2 Dec 2018

The authors present a methodology, with an accompanying R package, to emulate changes in crop yield under global change scenarios. The functions produced by this framework can be introduced in other models such as GCAM, which can help to speed-up different types of simulations. The manuscript is well written and presents important results that merit publication in GMD. However, I have one major comment to this work.

Although I really liked the Bayesian approach proposed here, which is more robust than previous linear regression approaches, I had problems understanding the approach for modeling the standard deviation term. It seems that the approach yields negative values of $\sigma_{CTW}$, which is evident by the use of absolute values in equation 6. As far as I am concerned, standard deviation values can never be negative since theoretically they are the square root of the variance. The choice of prior distributions for modeling

$\sigma_{CTW}$ expressed in equation 5, explains the reason for the negative values. For the baseline case, $b_0 \sim N(0, 0.001)$ yields a distribution of standard deviations centered around zero, which I find difficult to understand.

Modeling prior distributions for the variance in Bayesian analysis is not trivial, and there are many analyses dealing with this problem (e.g. see papers by Andrew Gelman). Most controversies about this topic deal with the choice of the prior distribution for variance parameters and whether gamma, inverse gamma, or other distributions are appropriate choices. These distributions however, are always defined in the positive part of the real line $\mathbb{R}^+$.

I suggest the authors to revise this part of the manuscript. If there is important information that I am missing regarding this issue, the authors should at least explain their choice of distribution and its interpretation.

---

## Author Comment (AC1) · 20 Feb 2019

**Review 1**
**Authors' Response:** We thank the reviewer for their thorough review of the paper. We have addressed every review point, to the improvement of the paper. Individual responses to different review points follow:

**Review:** I have two main concerns with the paper: the first refers to the C3MP dataset used to derive the emulator response functions, and the second refers to how these response functions are going to be used. The C3MP dataset is a large set of 99 CTW sensitivity tests carried out by a number of site-based crop models covering a range of different crops at a total of 1135 sites, of which data from 575 sites are

actually used in this study. The climate change signal used in these sensitivity tests is completely synthetic since it consists of applying a temporally uniform temperature offset or precipitation multiplier to a historical baseline weather timeseries. However, in reality, climate change is not constant over time. For example, precipitation might increase during part of the year, while decreasing during other times. I am not convinced that the constant CTW perturbation experiments are equivalent to using a more realistic climate timeseries, and I would suggest that the authors either show in the paper or at least point the reader to other literature showing that this is a valid approach. Otherwise, the authors risk that there is already a bias in the data used to train the emulator which would propagate to the emulated yields.

**Response:** We have clarified that we are looking at climatological mean TW changes during the growing season only, rather than changes to seasonality to our methods, as outlined in the two peer-reviewed citations detailing C3MP and its use. Analysis within Ruane et al., 2014 showed that the explicit modeling of future scenarios was quite consistent with aggregating the seasonal changes (see figure comparing simulated vs. emulated yields). Changes in seasonality will be reflected in the seasonal precipitation and temperature changes, but explicit action to adjust growing seasons to match new seasonality would require adaptation. Autonomous adaptation and technological gains are included in IAMs as part of the exogenous trend, so we focus on the primary climatological pressure, recognizing that we are excluding the secondary effects of seasonality. Climatological changes are considered here specifically because this is for use in GCAM, which solves for an equilibrium on 5 year timesteps during which subannual dynamics (such as the distribution of precipitation during the growing season) tend to average out. A user could investigate the impacts of different growing seasons by processing the climate data of interest from annual monthly to growing season average with an appropriate growing season mask for their intent. We believe such investigations are outside the scope of this paper. However, we have also included a subsection 2.1.1 explicitly addressing several known caveats of the C3MP data set.
**Review:** My second concern is that response functions for some of the 25 production groups are based on a very small number of sites, in the most extreme case only two sites. I cannot help but wonder how representative these results really are, keeping in mind that each production group represents one crop-irrigation-latitude combination, with latitudes only distinguished into extended tropics and mid-latitudes. In addition, these sites are not only used to derive a mean yield response, but also an estimate of response uncertainty. The high and low response functions are supposed to represent site responses at the mean plus/minus one standard deviation level but how meaningful are these estimates based on such a small sample size? The emulator response functions are only derived for two regions (extended tropics and mid-latitudes) but I wonder at what level of spatial detail they will be used later.

**Response:** We have included a new section 3.1.1 to directly address model performance in the production groups with small sample size. We also have clarified language regarding spatial scales throughout the paper. As a brief summary, we acknowledge that some crops and regions are under-represented in the training data. However, for many crops and regions, there is no comparable dataset available. In Section 3.1.1, we discuss validation that our modeling framework does represent the underlying data well for the production groups with small sample sizes. We also distinguish between the two-region response function and the degree of spatial heterogeneity that still results when combined with gridded temperature and precipitation change projections (throughout the text, but particularly in Section 2.1.1 Known caveats of the C3MP data set).

**Review:** In Figure 8 and 9 of the paper, the authors take spatial patterns of climate change from the HadGEM2-ES model and use their response functions to derive corresponding patterns of yield change. These patterns are shown, but not evaluated in any way. I would suggest that the authors compare their derived patterns of yield change to simulations of yield change from global gridded crop models for the same

climate data. After all, the intent of the emulator is to replace simulations by global gridded crop models. Such crop yield simulations for the HadGEM2-ES RCP8.5 scenario used in Figure 8 and 9 of the paper are, for example, available from a number of crop models and for a number of different crops within the ISIMIP data archive at https://esg.pik-potsdam.de/projects/isimip/ Such a comparison would help to address both of my concerns voiced above.

**Response:** We thank the reviewer for this helpful insight. We have added an extensive new section 4.1 directly comparing to the ISIMIP global gridded crop modeling results, as well as to other modeling efforts.

**Review:** Page 2, ll. 25 – 28: You state that previous emulators were restricted to emulating yield change under RCP scenarios. I am aware of at least two other global crop yield change emulators derived from crop model simulations that are applicable to any future climate scenario: Oyebamiji et al. (2015) and Ostberg et al. (2018). It might be useful to contrast them to the work presented in this paper.

**Response:** We apologize for this oversight and have added a discussion of these papers to our Introduction (paragraph beginning on P3L3).

**Review:** Page 4, starting with line 11, and Figure 1: You outline three use cases for the Persephone yield emulator, but none of this is actually done in this paper. So I'm not sure if the Methods section is the right place for this.

**Response:** We have streamlined this discussion of use cases and moved it from the Methods to the Introduction, as the intended use in GCAM has motivated many of our modeling choices.

**Review: Page 6, section 2.2:** The Copernicus guidelines request that datasets should be cited with a reference in the reference list. Is there a reference for the C3MP dataset?

**Response:** We have included language in Section 2.1 that the two peer-reviewed

C3MP publications (Ruane et al, McDermid et al) include data availability information. The relevant data to this work is also included in the paper analysis archive.

**Review:** Page 6, ll. 8 – 14: This part does not refer to the setup of C3MP or to your processing of the C3MP data. Instead, it refers to how climate data is pre-processed before use with the finished emulator. It should probably be moved to section 4.
**Response:** This has been done.

**Review:** Page 7, ll. 7 – 9: Unless I am mistaken, the global gridded crop models within AgMIP also conducted the 99 CTW sensitivity tests. They should offer a much better global coverage. Would it be worthwhile adding some discussion of why this paper used the site-based results instead of the global simulations?
**Response:** Unfortunately, the global gridded crop models did not participate in the 99 CTW tests that create the C3MP archive. The global gridded crop models have conducted their own, separate sensitivity tests and the data is not yet publicly available. Therefore, we developed this emulator using data we that was publicly available, but with the aim of a sufficiently flexible framework as to update to newer data sets (like the globally gridded crop models) if/when they become available. We have added language to the introduction and discussion highlighting that the Persephone framework described here can be updated in future versions, possibly with different explanatory variables, to include such gridded simulations as they become available. In the new section 4.1, comparing to the ISIMIP GGCM results, we now explicitly state that these models did not participate in the C3MP exercise

**Review:** Page 8, ll. 19 – 20: Are the 8 different functional forms documented anywhere? Are these the same as are used in the emulator?
**Response:** Thank you for highlighting this oversight. The functional forms used to estimate site-specific baseline Yields are now explicitly documented in the new

Appendix B.

**Review:** Page 11, ll. 9 – 10: Is the value of $b_0$ constrained between -0.02 and 0.02 or is it 0.02 %? The following sentence suggests that it is 0.02 %.
**Response:** We thank the reviewer for catching this typo. We have corrected this.

**Review:** Given the very small sample sizes of about 1/3 of the production groups the 84.135th and 15.865th percentiles do not seem very meaningful
**Response:** This is addressed with our new section 3.1.1 directly evaluating the performance of the three functional forms for small sample size production groups.

**Review:** Section 3.2: It seems to me that this is rather a test whether the crop models used in training feature these important relationships. Or would you say it is possible that these relationships are present in the models, but missing in the emulator? Since the evaluation is positive I guess it means that the relationships are present in the models and retained in the emulator
**Response:** Yes, the relationships are present in the models and retained by the emulator. The final sentence is correct and we have clarified the text to reflect this point.

**Review:** Page 20, l. 4 – page 21, l. 1: Climate change can affect both the start and the length of the growing season. Is this accounted for or is the same growing season used under climate change as during the reference period?
**Response:** The same growing season was used by every model for their training runs. We have updated our methods and discussion to explicitly state this. We also note that we plan to use with GCAM and are focused on long term climatological changes, rather than specific changes to seasonality that may average out over 5 year timesteps.

**Review:** Page 21, ll. 8 – 10: On the one hand you talk about passing CTW

changes for regions into the emulator, on the other hand you mention a gridded map of yield changes. So are the yield changes at the same spatial resolution as the climate data or is there a difference between resolutions (region versus grid)? Please clarify
**Response:** We have updated this language to be clearer (now occurs on P24).

**Review:** Page 21, ll. 17 – 18, and Figure 9: Looking at Figure 9, it seems to me that most regions show a positive yield change under the high response function, not just "a few regions". Also, I think that in the bottom row of Figure 9 the maps for mean and high response are swapped. The bottom right map (which should be the high response) looks identical to the map for Maize in Figure 8 (which shows the mean response).
**Response:** Thank you for catching this mistake, we have corrected the error in figure 9. Rather than swapping the mean and high response maps for Maize in figure 9, we inadvertently included the Wheat mean response map with the Maize high and low maps in the original figure. We have corrected the rainfed Maize mean figure, and clarified the language discussing Figure 9.

**Review:** Page 23, ll. 6 – 8: Here, you emphasize the rapid evaluation time of the response functions relative to a global gridded crop model, but I think you should really try to show that the emulator response is actually comparable to what you would get using a global gridded crop model. This step is missing in the paper.
**Response:** Thank you for this suggestion, we have added section 4.1, comparing many of our results to some of the ISIMIP GGCM results.

**Review:** Page 24, ll. 8 – 10: Given that the response functions are only derived for two latitudinal bands I would say that they cannot really be used to characterize the range of uncertainty within national or multi-national units (unless the respective unit is covered by C3MP sites).
**Response:** We have clarified the language in this section of the Conclusions and

discussion that the response functions are able to characterize the range of C3MP response sites, and that this is only a partial characterization of the response uncertainty within larger national land units. And that using more spatially complete data sets for training in the future will improve this characterization.

**Review:** Page 24: In the paragraph on caveats, I would suggest to add discussion of potential biases arising out of way the CTW experiments are set up. Another source of uncertainty results from the fact that crop model simulations generally omit adaptation options such as changing sowing dates or switching to different cultivars. This is done for simplicity and comparability but is not realistic considering that agriculture is a highly managed system, and potentially creates another bias in the crop model simulations of yield change that are used to train the emulator.
**Response:** We have added text to this effect.

**Review:** The greyscale lines in Figure 6, 8 and 9 are hard to see at all, let alone distinguish the different shades of grey.
**Response:** We have replaced the grayscale lines with black contours placed at values of 10

**Review 2**
**Review:**
The authors present a methodology, with an accompanying R package, to emulate changes in crop yield under global change scenarios. The functions produced by this framework can be introduced in other models such as GCAM, which can help to speedup different types of simulations. The manuscript is well written and presents important results that merit publication in GMD. However, I have one major comment to this work. Although I really liked the Bayesian approach proposed here, which is more robust than previous linear regression approaches, I had problems
understanding the approach for modeling the standard deviation term. It seems that the approach yields negative values of $\sigma$CTW , which is evident by the use of absolute values in equation 6. As far as I am concerned, standard deviation values can never be negative since theoretically they are the square root of the variance. The choice of prior distributions for modeling $\sigma$CTW expressed in equation 5, explains the reason for the negative values. For the baseline case, b0 âĹij N(0, 0.001) yields a distribution of standard deviations centered around zero, which I find difficult to understand. Modeling prior distributions for the variance in Bayesian analysis is not trivial, and there are many analyses dealing with this problem (e.g. see papers by Andrew Gelman). Most controversies about this topic deal with the choice of the prior distribution for variance parameters and whether gamma, inverse gamma, or other distributions are appropriate choices. These distributions however, are always defined in the positive part of the real line R +. I suggest the authors to revise this part of the manuscript. If there is important information that I am missing regarding this issue, the authors should at least explain their choice of distribution and its interpretation.

**Response:** We thank the reviewer for reading the work so closely. Indeed, the reviewer has correctly characterized our methodology, but highlighted that we did not communicate our methods clearly. We have clarified the text in our section on emulation, specifically addressing the reviewer's comments.

––––––––––––––––––––––––––––––